# Drought impact in the Bolivian Altiplano agriculture associated with El Niño—Southern Oscillation using satellite imagery data

Claudia Canedo-Rosso[1,2], Stefan Hochrainer-Stigler[3], Georg Pflug[3,4], Bruno Condori[5], and Ronny Berndtsson[1,6]

[1]Division of Water Resources Engineering, Lund University, P.O. Box 118, SE-22100 Lund, Sweden
[2] Instituto de Hidráulica e Hidrología, Universidad Mayor de San Andrés, Cotacota 30, La Paz, Bolivia
[3] International Institute for Applied Systems Analysis (IIASA), Schlossplatz 1, A-2361 Laxenburg, Austria
[4] Institute of Statistics and Operations Research, Faculty of Economics, University of Vienna, Oskar-Morgenstern-Platz 1, 1090 Wien, Austria
[5]Inter-American Institute for Cooperation on Agriculture (IICA), Defensores del Chaco 1997, La Paz, Bolivia.
[6] Center for Middle Eastern Studies, Lund University, P.O. Box 201, SE-22100 Lund, Sweden.

*Correspondence to*: Claudia Canedo-Rosso (canedo.clau@gmail.com)

**Abstract.** Drought is a major natural hazard in the Bolivian Altiplano that causes large agricultural losses. However, the drought effect on agriculture varies largely on a local scale due to diverse factors such as climatological and hydrological conditions, sensitivity of crop yield to water stress, and crop phenological stage among others, especially during a positive El Niño Southern Oscillation (ENSO) phase. To improve the knowledge of drought impact on agriculture, this study aims to classify drought severity using land surface data, analyse the relationship between drought and climate anomalies, and examine the spatio-temporal variability of drought using vegetation and climate data. Empirical data for drought assessment purposes in this area are scarce and spatially and unevenly distributed. Due to these limitations we used tested the performance of satellite imagery products for providing vegetation, land surface temperature (LST), and precipitation derived from satellite imagery, and gridded air temperature data products. Initially, we tested the performance of satellite precipitation and gridded air temperature data on a local level. With this information, t Then, the Normalized Difference Vegetation Index (NDVI) and LST were used to classify drought events, associated with past El Niño–Southern Oscillation (ENSO) phases. It was found that the most severe drought events generally occur during positive ENSO phase (El Niño years). In addition, wWe found that a decrease in vegetation is mainly driven by low precipitation and high temperature, and we identified areas where agricultural losses will be most pronounced under such conditions. The results show that droughts can be monitored using satellite imagery data when ground

data are scarce or of poor data quality. The results can be especially beneficial for emergency response operations and for enabling a pro-active approach to disaster risk management against droughts.

Keywords: Drought, agriculture, ENSO, NDVI, land surface temperature, climate variables, precipitation, and air temperature.

## 1.   Introduction

Agricultural production is highly sensitive to weather extremes, including droughts and heat waves. Losses due to such hazard events pose a significant challenge to farmers as well as governments worldwide (UNISDR, 2009, 2015). Worryingly, the scientific community predicts an amplification of these negative impacts due to future climate change (IPCC, 2013). Especially in developing countries such as Bolivia, drought is a major natural hazard

and Bolivia has experienced large socio-economic losses in the past due to such events (UNDP, 2011; Garcia and Alavi, 2018). However, the impacts vary on a seasonal and annual timescale, in regards to the hazard intensity, as well as the existing capacity to prevent and respond to droughts (UNISDR, 2009, 2015). Regarding the former, the El Niño Southern Oscillation (ENSO) plays an especially important role in several regions of the world, including the Bolivian Altiplano, as it drives losses of agricultural crops, and causes increased food insecurity

(Kogan and Guo, 2017). Most important rainfed crops in the region include quinoa and potato (Garcia et al., 2007). Generally speaking, agricultural productivity in the Bolivian Altiplano is low due to adverse weather and poor soil conditions (Garcia et al., 2003). On the other hand, low agricultural production levels can also be associated with the ENSO climate phenomena (Buxton et al., 2013).

The ENSO is a climate phenomenon that affects the precipitation variability of the Bolivian Altiplano (Thompson

et al., 1984; Aceituno, 1988; Vuille, 1999). The ENSO is defined as a periodical variation of the sea surface temperature over the tropical Pacific Ocean, and it represents neutral, warm (El Niño), and cold (La Niña) phases. The positive phase of ENSO is generally associated with warmer and dryer conditions, while the negative phase is associated with cooling and wetter conditions (Garreaud and Aceituno, 2001; Garreaud et al., 2003; Thibeault et al., 2012). ThusFor this area, droughts are generally driven by the positive ENSO warm phases in the study area

(Thompson et al., 1984; Garreaud and Aceituno, 2001; Vicente-Serrano et al., 2015). Previous research has addressed the influence of ENSO on agriculture in South America and the globe (see Iizumi et al., 2014; Ramirez-Rodrigues et al., 2014; Anderson et al., 2017). These studies were calling for a better understanding of the association between ENSO and agriculture to improve crop management practices and food security. However, to predict the ENSO effects is challenging, since the ENSO evolution depends not only on the tropic Pacific Ocean

temperature, but also on atmospheric convection, climate variability, and its teleconnection with other climate anomalies (Santoso et al., 2019).

The implementation of drought risk management approaches is now seen as fundamental (see e.g., the Sustainable Development Goals or the Sendai Framework for Risk Reduction) for sustainable development in vulnerable regions, including Latin American countries such as Bolivia (Verbist et al., 2016). To lessen the long-term impacts of these extreme events, the national government in Bolivia has taken several steps, e.g., to allocate budgets for emergency operations to compensate part of the losses occurred. Most of these measures are implemented ex-post (i.e., after a disaster event). However, based on ENSO forecasting, an El Niño event can be predicted 1 to 7 months ahead (Tippett et al., 2012) and consequently, there is an opportunity to implement additional ex-ante policies (i.e., before the event) to reduce societal impacts to droughts, increase preparedness, and generally improve current risk management strategies.

We embed our research within the IPCC framework (IPCC, 2012) and conceptually define disaster risk as a function of the hazard, exposure and vulnerability. Here, drought risk was defined as the likelihood of severe alterations in the normal functioning due to drought hazard interacting with the vulnerabilities of the exposed socio-environmental system, leading to potential adverse effects. Furthermore, disaster risk usually comprises different types of potential losses, sometimes very difficult to be quantified (UNISDR, 2009). In the case of drought, usually four different types are distinguished (UNISDR, 2009): meteorological-, hydrological-, agricultural- and socio-economic droughts. In more detail, a meteorological drought manifests if certain weather variables (e.g. precipitation) remain under predefined threshold levels over a certain time period while hydrological drought is usually determined through reduced water levels in water-bodies and ground water. Agricultural drought occurs when insufficient soil moisture and precipitation negatively affect crop yields while agricultural drought may turn into socio-economic drought if the supply or demand of agricultural products is negatively affected (see also the seminal paper of Wilhite and Glantz (1985)).

Using these concepts and definitions our research aims to: (1) classify agricultural drought severity (utilizing NDVI as a proxy for crop yields) using land surface data (our exposure component) and climate data (our hazard component), (2) analyse the relationship between drought and ENSO, and (3) assess drought through examining the spatio-temporal variability of vegetation, and its association with climate data (implicitly including the vulnerability component through the spatial-temporal variability). One major constraint for drought risk management in Bolivia is the scarce and uneven distribution of weather and agricultural production related ground data. To circumvent this problem, we test satellite-based and gridded data products (compared to available

empirically gauged data) to provide a full coverage (in respect to land area) for drought assessment and its spatial distribution across the region. Due to the particular importance of ENSO for drought risk management, we additionally assess the impacts associated with ENSO on agriculture for the Bolivian Altiplano. Furthermore, we give indications what climate variables may be most important in which regions to predict drought losses that can

further be used for hotspot selection. The paper is organized as follows, section 2 provides a discussion of the data used, while section 3 present ~~presents the~~the methodology applied ~~and data used~~. Afterwards,~~, and~~ section ~~3~~4 presents the corresponding results found and section 5 ends with a conclusion and outlook to the future.~~. Section 4 puts the results into a context of drought impact and hotspot selection with conclusion.~~

## 2.   Data Used ~~and Methodology~~

### 2.1 ~~Ground data and satellite imagery~~Climate data

~~The~~ Our methodology ~~applied~~ is very much related to the data scarce situation for the Bolivian Altiplano and we therefore start with an introduction of the available datasets that ~~we~~are used for our purpose~~s~~. In regard~~s~~ to the climate dimension, the Altiplano has a pronounced southwest-northeast precipitation gradient (200–900 mm year$^{-1}$) during the wet season occurring from November to March (Garreaud et al., 2003). Over 70% of total

precipitation occur~~s~~ during the summer months (from December to February, see Fig. 1a) in association with the South American Monsoon (see Zhou and Lau, 1998; Garreaud et al., 2003). Time series of monthly precipitation at 12 locations as well as mean, maximum, and minimum temperature at 8 locations from September 1981 to August 2015 were available from the National Service of Meteorology and Hydrology (SENAMHI) of Bolivia (see Table A1). These data sets ha~~ve~~d less than 10% of missing data and therefore served well for our analysis. ~~.~~

As already indicated, precipitation and temperature gauge locations are unevenly distributed and mainly concentrated in the northern Bolivian Altiplano. To improve the spatial coverage of climate related data, monthly quasi-rainfall time series from satellite data the Climate Hazards Group InfraRed Precipitation with station data (CHIRPS) were included in our study. CHIRPS represents a 0.05° spatial resolution satellite imagery and a quasi-global rainfall dataset from 1981 to the near present (Funk et al., 2015). The advantage of using CHIRPS is the

high spatial resolution of data, obtained with resampling of TMPA 3B42 (with 0.25° grid cell). The spatial resolution represents a better option for agricultural studies as well and therefore is most appropriate for our approach (CHIRPS is described in detailed at http://chg.geog.ucsb.edu/data/chirps/).

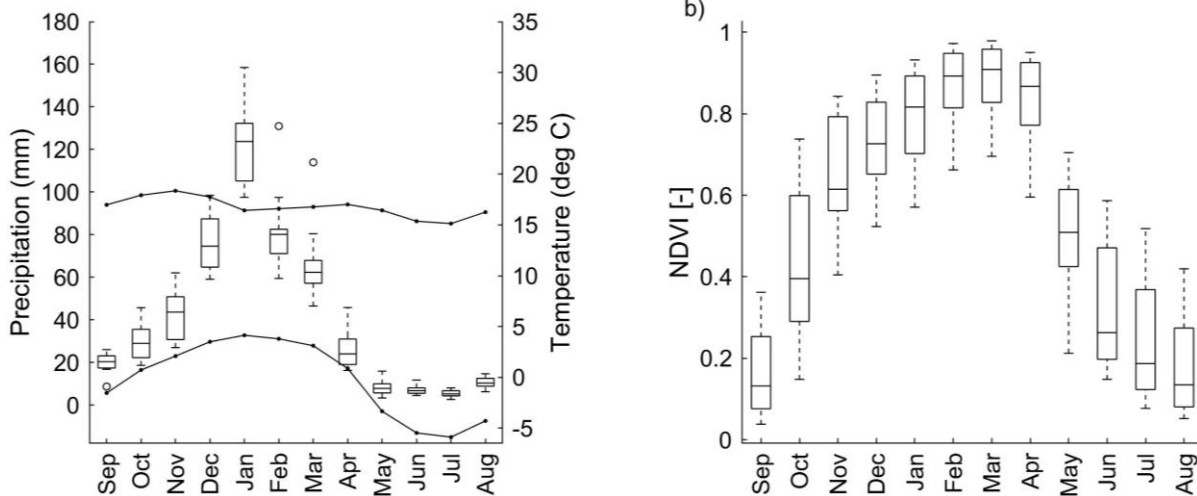

Fig. 1. (a) Gauged mean monthly total precipitation and average maximum and minimum temperature from September 1981 to August 2015. (b) Mean monthly NDVI at the same spatial locations. Lower and upper box boundaries 25th (Q1) and 75th (Q3) percentiles, respectively, line inside box is median, lower and upper error lines 1.5 times the interquartile range (Q3-Q1) from the top or bottom of the box, white circles data falling outside 1.5 times the interquartile rage.

Additionally, ~~satellite~~ a gridded dataset of monthly mean air temperature was obtained from the Physical Sciences Division (PSD) of the US National Oceanic and Atmospheric Administration (NOAA, https://www.esrl.noaa.gov/psd/) defined by Willmott and Matsuura~~.~~, using a spatial interpolation of ~~a~~ composite
5 stations records from the Global Historical Climatology Network (GHCN version 2) and Legates and Willmott's (Legates and Willmott, 1990b; Legates and Willmott, 1990a). The ~~satellite~~ gridded air temperature dataset has a resolution of 0.5° by 0.5° and was available during the study period from September 1981 to August 2015. This dataset incorporates station-height information, through an average air-temperature lapse rate (Willmott and Matsuura, 1995). Here, a digital-elevation-model was used for the interpolation to adjust air temperatures in relation to ~~the~~ sea level.

### 2.2 Land surface data

Apart from climate datasets, NDVI was assembled from the Advanced Very High Resolution Radiometer (AVHRR) sensors by the Global Inventory Monitoring and Modelling System (GIMMS) at semi-monthly (15 days) time steps with a spatial resolution of 0.08°. NDVI 3g.v1 (third generation GIMMS NDVI from AVHRR

sensors) was available from September 1981 to August 2015. The NDVI is an index that presents a range of values from 0 to 1, bare soil values are closer to 0, while dense vegetation is close to 1 (Holben, 1986). NDVI 3g.v1 GIMMS provides information to differentiate valid values from possible errors due to snow, cloud, and interpolation. These errors were removed from the dataset and replaced with the nearest neighbouring value.

Additionally, the monthly Land Surface Temperature (LST) was obtained from the Global Land Data Assimilation System (GLDAS) by the Noah Land Surface Model L4 monthly version 2.0. ~~The LST dataset has a resolution of 0.25° and it~~ The dataset was available for the study period ~~from~~ September 1981 to August 2015. The LST estimations from GLDAS were based on remotely sensed observations of AVHRR (Rodell et al., 2004) and include an algorithm that relies on an optimal interpolation routine (Ottlé and Vidal-Madjar, 1992) to

assimilat~~ing~~e the LST onto a 0.25 to 0.25 degree grid. This data record was selected due to its temporal resolution, however it is important to mention that a higher spatial resolution could improve the accuracy of agricultural analyses, and further reduce the uncertainties of the data noise originating from land heterogeneity.

~~This~~Finally, the study was conducted for the ~~A~~agricultural land in the Bolivian Altiplano ~~covers about~~ (~200,000 km$^2$). ~~, and it~~ The agricultural land was spatially identified based on the land use map developed by the

Autonomous Authority of the Lake Titicaca (for the northern Altiplano) in 1995 at a scale of 1:250,000 (UNEP, 1996), and the Ministry of Development Planning in 2002 using Landsat imagery and ground information at a scale 1:1,000,000 (geo.gob.bo, for the southern Altiplano).

## 3.   Methodolog~~y~~s

The analysis of drought impact on agriculture for the Bolivian Altiplano and its relationship with the ENSO is

based on the following three steps. Firstly, an evaluation of satellite precipitation and gridded air temperature against gauged datasets was performed to investigate the accuracy of these estimates compared to empirical on-the-ground date. Secondly, the severity of drought was classified using land surface data, and using this information drought events were associated with the ENSO variability. Finally, a stepwise regression approach was used to study the variability of vegetation and its relationship with corresponding climate variables. The

overall aim of our study is to investigate drought effects on agriculture through the analysis of land surface and climate variations, and its relation with the ENSO anomalies.

## 3.1 ~~Validation of satellite-based data products~~Evaluation of climate data

The performance of the satellite-based data (compared to empirical ground data, see Fig. 2) to accurately estimate the amount of rainfall (for example for ~~to assess~~ rain detection capability purposes) was based on statistical measures for monthly pair-wise time series, including categorical analyses, and follows ~~the~~ methodologies suggested and~~y~~ applied in previous studies in this region, ~~(~~and were selected for comparison reasons (Blacutt et al., 2015; Satgé et al., 2016). The mean error (ME), bias, and mean absolute error (MAE) were calculated based on Wilks (2006). These measures evaluate the prediction accuracy of the satellite data compared to gauged data. The ME and bias show the degree of over- or underestimation (Duan et al., 2015). In contrast, as measuring the absolute deviation, MAE shows only non-negative values. The ME, bias, and MAE perfect match correspond to zero between gauge observation and satellite-based estimate. Furthermore, and similar to Blacutt et al. (2015) and Satgé et al. (2016), the Spearman's rank correlation was computed to estimate the goodness of fit to observations. To evaluate results, and in accordance to~~as in~~ similar studies, correlation coefficients larger or equal to 0.7 were considered as reliable (Condom et al., 2011; Satgé et al., 2016). The ME, bias, and MAE were calculated, respectively according to Eqn. (1), (2), and (3) (Table 1).

Table 1. Accuracy measures for climate estimation data ~~satellite data~~ performance evaluation. Here, N is the number of samples, $S_i$ is the climate estimation~~satellite-based dataset~~ for month $i$, and $G_i$ is the gauged dataset for the same month. H is a hit, F is a false alarm, and M is a miss.

| Statistical indicator | Abbreviation | Units | Equation | |
|---|---|---|---|---|
| Mean error | ME | mm, °C | $\sum(S_i - G_i) / N$ | (1) |
| Bias | Bias | % | $\sum(S_i - G_i) / \sum G_i \times 100$ | (2) |
| Mean absolute error | MAE | % | $\sum |(S_i - G_i) / G_i| / N \times 100$ | (3) |
| Probability of detection | POD | - | $H / (H + M)$ | (4) |
| False alarm ratio | FAR | - | $F / (H + F)$ | (5) |

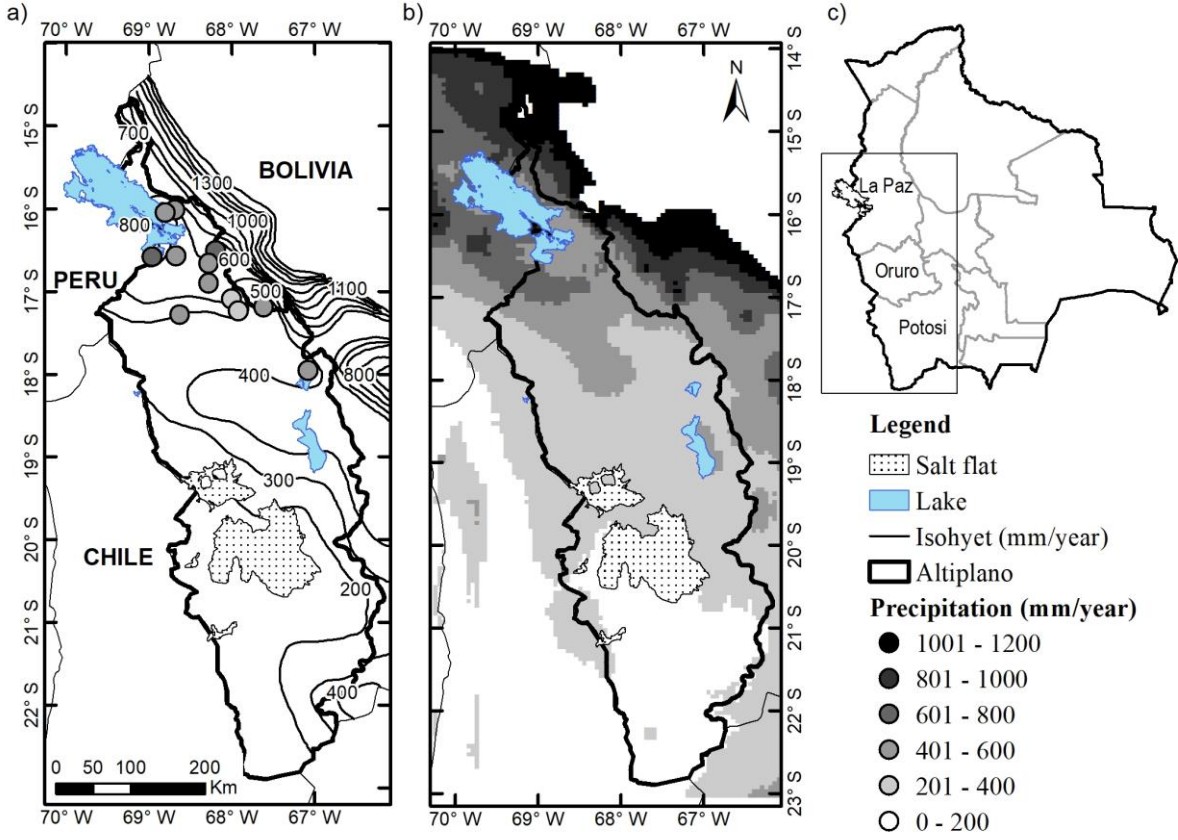

Fig. 2. Mean of total annual precipitation from September 1981 to August 2015 for: (a) gauged precipitation data (circles) and isohyets (solid line), (b) the CHIRPS satellite rainfall product, and (c) Bolivia, and the major political divisions of the Bolivian Altiplano: La Paz, Oruro and Potosi.

Two statistical indicators based on a contingency tables were computed for the categorical statistics, namely Probability of Detection (POD) and False Alarm Ratio (FAR). The POD indicates what fraction of the observed events that was correctly estimated, and FAR indicates the fraction of the predicted events that did not occur (Bartholmes et al., 2009; Ochoa et al., 2014; Satgé et al., 2016). The POD and FAR range from 0 to1, where 1 is a perfect score for POD, and 0 is a perfect score for FAR. These measures were used to evaluate the satellite precipitation and the gridded air temperature estimations. Here, the rainfall amounts are considered as binary values, i.e., rain occurrence or absence. Based on this approach, three counting variables were taken into account: the number of events when the satellite rain estimation and the rain gauge report a rain event (hit or H), when only

the satellite reports a rain event but no rain on the ground is observed (false alarm or F), and when only the rain gauge reports a rain event but not the satellite and therefore is a miss (M). The POD and FAR were calculated, respectively according to Eqn. (4) and (5) (Table 1).

Besides the precipitation data, ~~satellite~~ the gridded air temperature data were ~~validated~~ evaluated using ground data. The ~~satellite~~ gridded air temperature was correlated with the mean gauged temperature at the same spatial location. The mean temperature of the gauged data was calculated using the arithmetic mean between the maximum and minimum temperature. The regression performance was evaluated using the monthly pair wise time series to define the Spearman's rank correlation, relative ME, bias, and MAE. For air temperature, the MAE was defined using absolute values.

## 3.2 Drought associated with ENSO

Healthy vegetation usually shows enlarged near infrared and reduced visible red band, and emits less absorbed thermal infrared radiation ~~shows a~~ resulting in lower surface temperature ~~due to the absorption of thermal infrared radiation~~ (Kogan and Guo, 2017). Therefore, vegetation indices and land surface temperature (LST) are widely used for water and energy balance approaches (see Moran et al., 1994; Corbari et al., 2010; Sánchez et al., 2012; Helman et al., 2015). Previous findings indicate a negative (positive) relationship between LST and NDVI caused by limited moisture (energy-temperature) availability for vegetation growth (Karnieli et al., 2010). Drought spells typically present low NDVI and high LST due to vegetation deterioration and higher contribution of the soil signal (Kogan, 2000). Here, we study the relationship between LST and NDVI using the Vegetation Health Index (VHI, Eqn. (8)) developed by Kogan (1995) that combines the Vegetation Condition Index (VCI, Eqn. (6)) and Temperature Condition Index (TCI, Eqn. (7)). VCI is a normalized NDVI that allows to seek the variability of the signal, showing an increased VCI when NDVI increases. (Kogan, 1995; Kogan, 2000; Kogan and Guo, 2017). In contrast, the TCI formulates a reverse ratio compared to the VCI, decreasing when LST increases, assuming that higher land surface temperatures suggest a decreasing soil moisture causing stress of the vegetation canopy.

Table 2. Drought classification indices.

| Drought index | Acronym | Equation | |
|---|---|---|---|
| Vegetation Condition Index | VCI | $(NDVI_i - NDVI_{min}) / (NDVI_{max} - NDVI_{min})$ | (6) |
| Temperature Condition Index | TCI | $(LST_{max} - LST_i) / (LST_{max} - LST_{min})$ | (7) |
| Vegetation Health Index | VHI | $0.5\ VCI + 0.5\ TCI$ | (8) |

where NDVI$_i$, NDVI$_{max}$ and NDVI$_{min}$ (LST, LST$_{max}$ and LST$_{min}$) are monthly NDVI (LST) and the month absolute maximum and minimum from September 1981 to August 2015, respectively. We took a mean of VCI and TCI assuming that they equally contribute to the VHI.

The VCI, TCI, and VHI was defined for each month during the growing season (from September to April). We assumed the occurrence of a drought event when the indices were lower than 40%. The classification of drought was established based on the severity of the event in which five classes were defined: extreme ($\leq 10$), severe, ($\leq 20$), moderate ($\leq 30$), mild ($\leq 40$), and no ($>40$) drought (Bhuiyan and Kogan, 2010).

The drought events were further classified based on the occurrence of El Niño and La Niña events (Table 3). The classification ENSO was obtained from Null (2018). El Niño and La Niña events were identified from 5 consecutive overlapping 3-month mean sea surface temperature for the Niño 3.4 region (in the tropical Pacific Ocean). A moderate El Niño (La Niña) was defined as 5 consecutive overlapping 3-month periods at or above the +1.0 ° to + 1.4 °C anomaly (-1.0 ° to -1.4 °C), strong El Niño (La Niña) event for a threshold between +1.5 ° to +1.9 °C anomaly (-1.5 ° to -1.9 °C anomaly), and a very strong El Niño event for a threshold equal or greater than +2 °C anomaly (https://ggweather.com/enso/oni.htm). For this study, a neutral or weak phase was defined as a threshold between -0.9 ° to +0.9 °C anomaly.

Table 3. El Niño and La Niña phases (from Null (2018)).

| El Niño | | | La Niña | |
| --- | --- | --- | --- | --- |
| **Moderate** | **Strong** | **Very Strong** | **Moderate** | **Strong** |
| 1986-87 | 1987-88 | 1982-83 | 1995-96 | 1988-89 |
| 1994-95 | 1991-92 | 1997-98 | 2011-12 | 1998-99 |
| 2002-03 | | 2015-16 | | 1999-00 |
| 2009-10 | | | | 2007-08 |
| | | | | 2010-11 |

### 3.3 Regression of vegetation and climate variables

A stepwise regression approach was used to quantify the dependency between vegetation and ~~satellite based~~ climate variables (satellite-based precipitation and gridded air temperature; Eqn. 10) ~~further~~ to be used for the hotspot selection process. In more detail, the results presented here are a combination of forward and backward selection techniques to increase the robustness of the results (in terms of explanatory power, i.e., variability explained, as well as variable selection, i.e., same variable selected across a range of possible models). The

independent variable considered was NDVI, and the dependent variables were selected to include precipitation and air temperature (for the same spatial location across the study region). We assumed that NDVI represents the crop phenological stages of the growing season that is from September to April (Fig. 1). Precipitation was selected as predictor due to its relevance for water availability for vegetation growth. Precipitation is the main source of water in the Altiplano because only 9% of the Bolivian cropped surface area are irrigated (INE, 2015). Air temperature is a relevant variable due to photosynthetic and respiration processes (Karnieli et al., 2010). Firstly, the NDVI was related to CHIRPS rainfall datasets. Secondly, air temperature was included in the analysis. HereFor this, only the NDVI grids for agricultural land were selected. Since, agricultural production data are scarce in the region, we suggest that crop yield data can be improved using the NDVI. Besides improving the crop yield resolution, the NDVI also allows to analyse the variability of vegetation at a monthly time scale. This makes it possible to analyse the phenology of the studied crops through to the growth phases. NDVI estimates the vegetation vigour (Ji and Peters, 2003) and crop phenology (Beck et al., 2006). The final regression model for each spatial unite therefore wasis defined as

$$NDVI = \beta_0 + \beta_1 \, precipitation + \beta_2 air \; temperature \qquad (10)$$

For the forward selection, the variables were entered into the model one at a time in an order determined by the strength of their correlation with the criterion variable (only including variables if they present a confidence level of 95%). The effect of adding each variable was assessed during its entering stage, and variables that did not significantly add to the fit of the model were excluded (Kutner et al., 2004). For backward selection, all predictor variables were entered into the model first. The weakest predictor variable was then removed and the regression fit re-calculated. If this significantly weakened the model then the predictor variable was re-entered, otherwise it was deleted. This procedure was repeated until only useful predictor variables (in a statistical sense, e.g., significant as well as model fit) remained in the model (Rencher, 1995). The results were compared with results from literature regarding phenology and weather-related characteristics of crops.

It should be noted that the precipitation in the Altiplano shows a marked rainy season from November to March. The peak of precipitation is in December and January (Fig. 1a). AndAdditionally, NDVI displays a peak in March and April (Fig. 1b). The lag between the precipitation and NDVI is reasonable since vegetation requires time to grow (e.g. Shinoda, 1995; Cui and Shi, 2010; Chuai et al., 2013). Considering this lag-time, the 3-month time series of NDVI was regressed with the 3-month time series of the climate variables (satellite-based data products of precipitation and gridded air temperature) during the growing period for the agricultural land. First, the NDVI and the climate variables were related considering the overlappinged 3-month time series, and afterwards a

relation was developed considering a lag from 1 to 4 months between NDVI and climate variables, resulting in 22 regressions per NDVI grid. The regressions were developed for each NDVI grid separately, associated with the nearest precipitation and air temperature dataset. Previous to the stepwise regression analysis, the 3-month time series of NDVI, satellite precipitation and ~~satellite~~ gridded air temperature data were standardized.

**3.4. Results**

Validation of the satellite rain data using empirical precipitation data from the weather stations was done for the 12 locations where gauge precipitation data were available (see Fig. 2 and Table A1). The qualitative methods discussed in section 2.2 for the CHIRPS rainfall estimates show differences between summer (from December to March) and winter season (from June to August). CHIRPS data show better accuracy during summer. The
precipitation during the austral summer is highly relevant because it concentrates the 70% of the annual rainfall (Garreaud et al., 2003) and it occurs during the growing season. During May, CHIRPS data show lower accuracy compared to the other months. The precipitation from May to August is almost null in the study area (Fig. 1) and it will be further described as the dry season. This season presents stable atmospheric conditions with few precipitation events (Garreaud et al., 2003).

Interestingly, the spearman rank correlation between monthly gauged precipitation and satellite rain product datasets was significant (p-value <0.05) for all locations. The correlation coefficients (r) vary from 0.5 to 0.8 (mean = 0.7). The ME and bias disclose an underestimation of precipitation estimation during October, November, and April, and an overestimation during the summer season (mean = 5 mm and 7%, respectively) with a peak in February. For the MAE coefficient, CHIRPS estimations are more accurate during the rainy season (mean = 31%).
In contrast, CHIRPS data indicate poor accuracy during the dry season (mean MAE = 92%). From June to August, CHIRPS data present an underestimation of the gauged precipitation (mean bias = -39%). Summarizing these observations, we conclude that the CHIRPS-rainfall dataset is more accurate during the rainy season, and it represents an adequate alternative in case of lack of gauged data or in case of poor data quality. However, it should be noted that such data still must be used with caution considering the uncertainties due to the under or
overestimation of precipitation along the heterogeneous topography of the Altiplano (see Paredes-Trejo et al., 2016; Paredes-Trejo et al., 2017; Rivera et al., 2018).

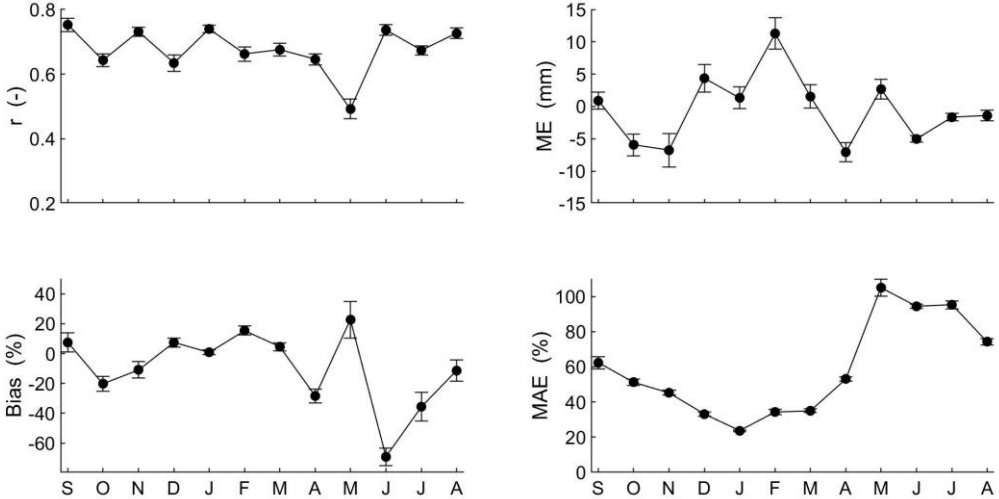

Fig. 3. Monthly accuracy measures of CHIRPS-rainfall data product. Mean monthly values are represented by black circles, and bars represent the standard error of the mean.

Moving from rainfall to temperature, the inter-annual temperature at the 8 locations varied considerably between
5    summer (from December to March) and winter (from June to August), including a larger variance for the minimum temperature (Fig. 1a). The mean monthly air temperature from ~~satellite~~ gridded data was compared with mean temperature of gauged data. The ~~satellite~~ gridded air temperature underestimated the mean gauged temperature, and this error could be due to the high elevation and cloud coverage. The spearman correlation at the 8 stations displayed coefficients from 0.1 to 0.7. From November to April, the gridded air temperature ~~satellite based~~
10    ~~estimations~~ show significant correlations (p-value <0.05). Large correlations are shown during summer season (mean = 0.7), while the other months show rather weak correlations. ME and bias show a slight underestimation from October to April (mean = -0.5 ºC and -4% respectively), and an overestimation from May to August (mean = 0.3 and 12% respectively). Finally, MAE is about 1.2 ºC~~0%~~ from September to April, higher values develop during winter season (mean = 1.6 ºC~~32%~~). In conclusion, the ~~satellite~~ gridded air temperature data product
15    performs better from November to April. Similar to the precipitation data, the application of ~~satellite~~ gridded air temperature data must take into account the potential errors due to the estimation uncertainties, mainly during winter season.

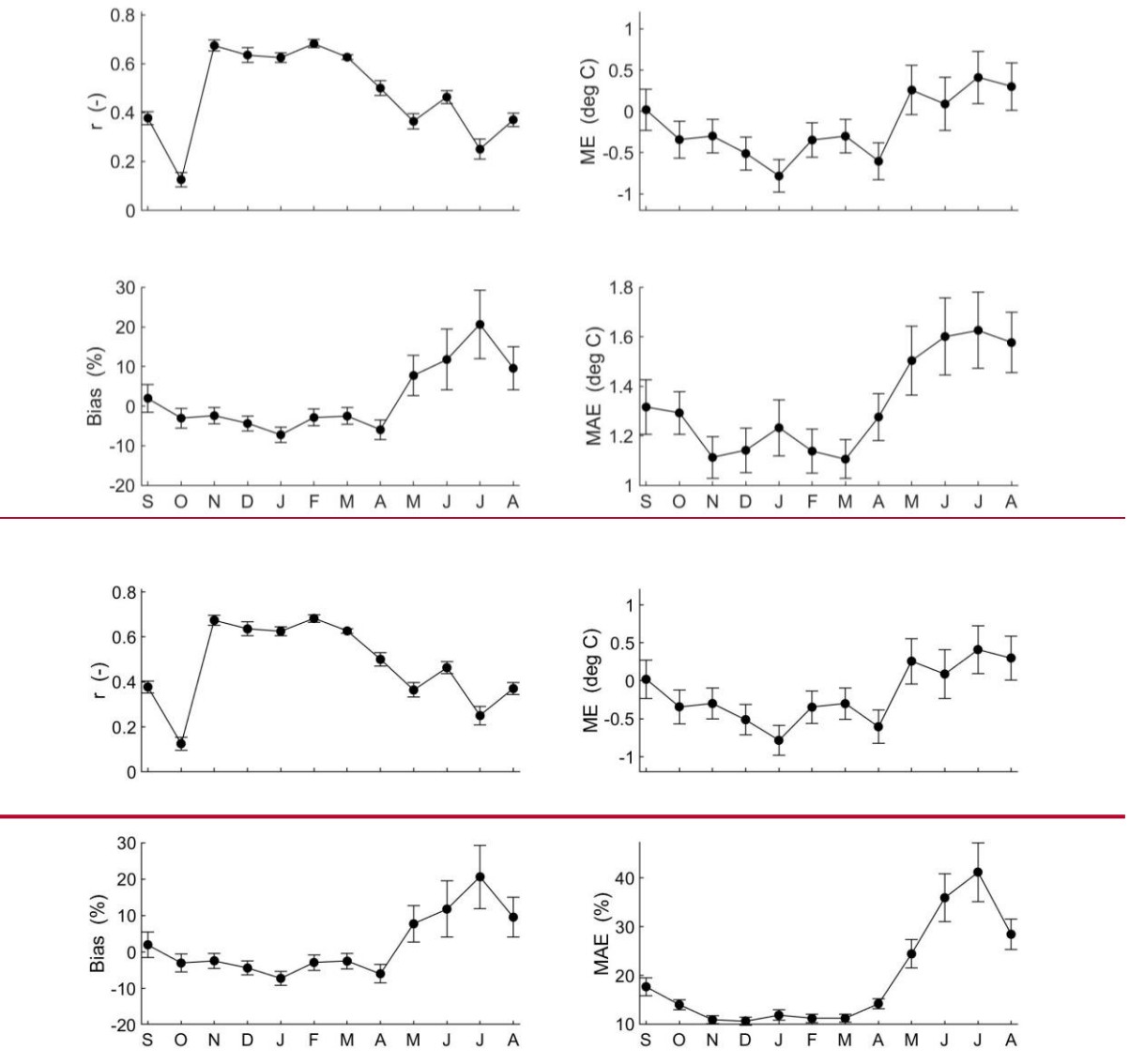

Fig. 4. Same as Fig. 3 but for accuracy measures of ~~satellite based~~gridded air temperature data product.

As discussed above, the VCI, TCI, and VHI were calculated during the growing season. The sowing period

5    depends on the initial soil moisture content, therefore the beginning of the growing season oscillates from

September to November (Garcia et al., 2015). For this reason, the drought severity was classified considering the

mean of VCI, TCI, and VHI for the agricultural land during November-April. Figure A1 shows mean monthly

VCI from November 1981 to April 2015. The major drought events (severe or extreme) are visible in 1982-83,

1983-84, and 2009-10. Followed by moderate drought events during 1987-88, and 1993-94, and several mild events. Figure A2 shows the mean monthly TCI, where the major drought events (severe or extreme) occurred in 1982-83, 1987-88, 1997-98, 2004-05, and 2009-10. Followed by moderate drought events during 1981-82, 1983-84, 1994-95, 2006-07, and 2008-09, and several mild events as well. Finally, Fig. A3 shows the VHI results, in
which the major drought events occurred during 1982-83, 2004-05, and 2009-10.

In a next step~~Further,~~ we related drought indices with the ENSO phases (Table 4). Extreme, and severe droughts were generally found during El Niño phase. The extreme drought of 1982-83, coincided with a very strong El Niño phase. For this event, the largest economic losses caused by droughts during the study period are~~were~~ reported (Table 5). Followed by the very strong El Niño phase of 1997-98, which reported the second largest
economic losses. Besides these two main drought events, the strong El Niño~~no~~ 1987-88 coincided with an extreme/moderate drought (TCI≤10%, VCI≤30%) classification. During this period, large economic losses were reported as well (Table 5). In contrast, the strong El Niño 1991-92 showed low severity (mild drought VCI≤40%), and no economic losses were reported. This indicates that despite El Niño phenomenon is generally associated with drought in the Altiplano, there are several other mechanisms that drive a drought occurrence and determine
its severity. For instance, dry (wet) and warm (cool) conditions during El Niño (La Niña) phases are generally shown in the tropics (Garreaud et al., 2003). However, an anomalous location and intensity of zonal wind anomalies could cause disturbances of the warming and cooling air patterns causing rainfall anomalies on the Altiplano (Garreaud and Aceituno, 2001). This is the case of the dry La Niña 1988-89 that showed a mild drought classification (TCI≤40%).

Table 4. Drought indices classification during ENSO phases.

| ENSO | Drought | VCI | TCI | VHI |
|------|---------|-----|-----|-----|
| El Niño | Extreme | | 1982-83, 1987-88, 1997-98 | |
| | Severe | 1982-83, 2009-10 | 2009-10 | 1982-83, 2009-10 |
| | Moderate | 1987-88 | 1994-95 | |
| | Mild | 1986-87, 1991-92 | 1986-87 | 1994-95, 1997-98 |
| La Niña | Mild | 1995-96, 2007-08, 2010-11 | 1988-89 | |
| Neutral/ weak | Extreme | | 2004-05 | |
| | Severe | 1983-84 | | |
| | Moderate | 1993-94 | 1981-82, 1983-84, 2006-07, 2008-09 | 2004-05 |

| Mild | 1981-82, 1996-97, 2003-04, 2008-09 | 1984-85 1990-91 1993-94 2014-15 | 1981-82, 1983-84, 1990-91, 1993-94, 2005-06, 2008-09 |

One severe (1983-84) and one extreme (2004-05) event occurred during a neutral/weak ENSO. The severe drought (VCI ≤ 20%) occurred during a neutral phase of 1983-84. This coincides with the findings of Vicente-Serrano et al. (2015), that analyzed the standardized precipitation/evaporation index in Bolivia, which is an alternative technique to characterize a meteorological drought. The extreme drought (TCI ≤ 10%) of 2004-05 occurred in November and December. From January to April of 2004-05 the VCI and VHI were above 40%, and there were no claims of drought losses in the Altiplano for this particular year (Table 5). Besides these two events, moderate and mild droughts also occurred during non El Niño phases.

Table 5 shows that five drought events were reported during a neutral ENSO phase. In 2012-13, the largest impact occurred, affecting about 80 000 people in the Altiplano (Desinventar, 2020). Despite that the mean of the drought indices indicates no drought during this period (VCI, TCI, and VHI >40%), some spatial locations in the study region indicated the occurrence of a drought event in November and December (21% and 29% of the total studied grids showed mild and moderate droughts for the TCI and VCI respectively).

Table 5. Drought impact in Bolivia (from EM-DAT (2020), BID (2016), and CAF (2000)).

| Year | ENSO phase | Affected people | Total damage ('000 US$) |
| --- | --- | --- | --- |
| 1982-83 | El Niño | 3 083 049 | 917 200 |
| 1987-88 | El Niño | | 48 400 |
| 1989-90 | Neutral | 283 160 | |
| 1997-98 | El Niño | | 279 310 |
| 1993-94 | Neutral | 50 000 | |
| 1999-00 | La Niña | 20 000 | |
| 2003-04 | Neutral | 55 000 | |
| 2007-08 | La Niña | 27 500 | |
| 2009-10 | El Niño | 62 500 | 100 000 |
| 2012-13 | Neutral | 340 355 | |
| 2013-14 | Neutral | 51 180 | |

Regarding the relationship between vegetation and climate variables, we note that the precipitation season occurs mainly during the austral summer months (from December to March), and the vegetation development shows a

lag with a maximum development of about March and April (Fig. 1). The NDVI (Fig. 1b) shows a similar growing pattern as the crop phenology in the region, which starts in September and ends in April. Maximum and minimum temperature varies during the year. Higher temperature during the austral summer leads to higher evapotranspiration and a decrease of water retained in the root zone. With this presumption, stepwise linear regression models were tested using 3-month time series of NDVI as dependent variable and 3-month time series of satellite-based data product of precipitation and gridded air temperature as independent variables (Eqn. (10)). The stepwise regression was defined considering the overlapped 3-month time series, and the 3-month time series with a lag from 1 to 4 months at the same spatial location over the agricultural land.

The results of the stepwise regression show larger coefficient of determination ($R^2$) in the northern and central Bolivian Altiplano, starting from the southern Lake Titicaca and moving southwards to the Lake Poopó, and close to the rivers paths. Lower $R^2$ a ires shown along the southwestern Bolivian Altiplano, that could be explained through the large variance of the NDVI, which may depend to on other factors besides precipitation and temperature, including crop management. Figure 5 shows the $R^2$ of the best fit regression in the Bolivian Altiplano for the three-month period of NDVI and the climate variables (precipitation and temperature) during the beginning and end of the growing season. It can be seen that the NDVI depends largely on the studied climate variables. This may be due to the crop´s sensitivity for water stress during specific stages of the growing season. For instance the most sensitive stages of the quinoa crop are the emergence, flowering, and grain development (see Geerts et al., 2008; Geerts et al., 2009), and the near absence of irrigation practices in most of these regions.

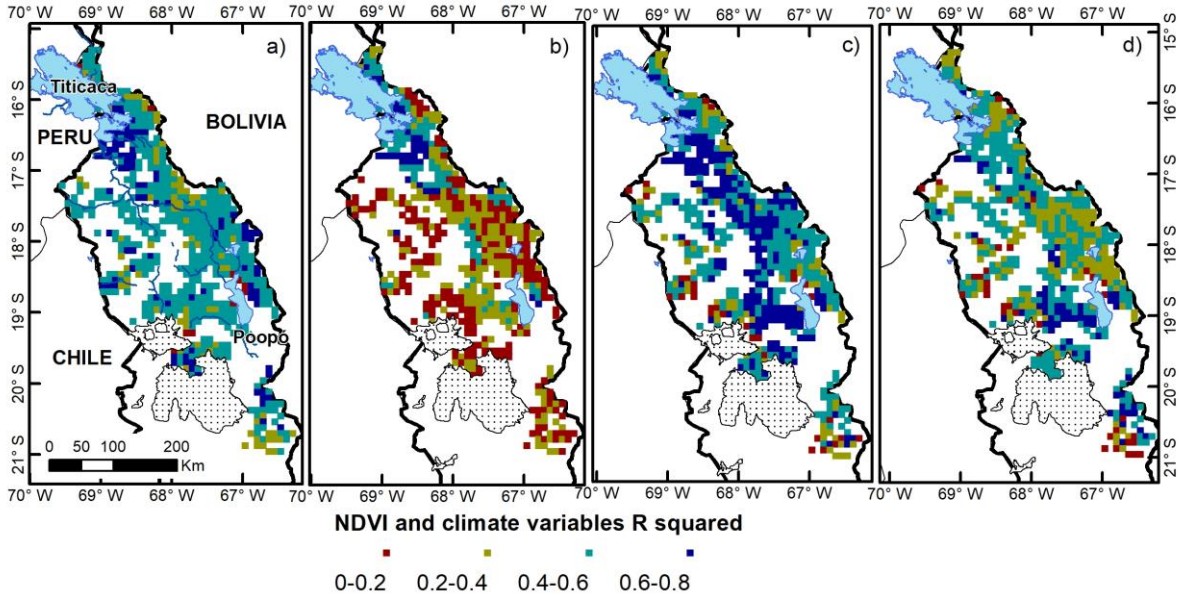

Fig. 5. Coefficient of determination ($R^2$) of NDVI for the 3-month time series for a) SON, b) OND, c) MAM and d) MAM and the climate variables (satellite precipitation and ~~gridded~~ air temperature products) for SON, SON, FMA, and MAM respectively. The significant regression coefficients for precipitation (air temperature) cover: a) 45% (98%), b) 64% (91%), c) 95% (96%), and d) 23% (98%) of the total studied grids that represent the agricultural land.

In more detail, the stepwise regression results for the overlapping 3-month time series of NDVI and climate variables for SON (September, October, and November) show statistically significant coefficients for precipitation and air temperature at 45% and 98% the agricultural area in the Bolivian Altiplano with a median of 0.2 and 0.7, respectively (Fig. 5a). This indicates that the NDVI increases with more rain and higher air temperature. Interestingly, the significant regression coefficients of NDVI for OND (October, November, and December) associated with precipitation and air temperature for SON cover 64% and 91% of the agricultural area, and have a positive median of 0.3 and 0.4, respectively (Fig. 5b). A time-lag of one month shows larger spatial coverage of response of vegetation to precipitation anomalies. Here, the largest coefficient of determination are shown in areas surrounding the Lake Titicaca. Moreover, the response of the NDVI for MAM (March, April, and May) to the studied climate anomalies for FMA (February, March and April) covers 95% and 96% of the agricultural land for precipitation and air temperature, respectively (Fig. 5c). This mostly shows coefficients of determination ranging from 0.4 to 0.8, and positive regression coefficients for precipitation and air temperature have a median of 0.5 and 0.4, respectively. The hours of sun required for crop development could be an~~the~~ explanation for the time-lag

between vegetation and the climate variables. In addition, the lag differences between vegetation and precipitation can be partly explained by the topography, land cover, ground-water, and soil properties (Yarleque et al., 2016). Finally, the regression for NDVI and climate variables for the overlapped 3-month time series of MAM shows significant coefficients at 23% and 98% of the agricultural land, with a median of 0.4 and 0.6 for precipitation and

air temperature, respectively (Fig. 5d). Hence, the vegetation response to precipitation is limited for the last overlapped 3-month time series of the growing season. However, it should be noted that air temperature remains an important variable.

To summarize, while acknowledging some important limitations, we found that the CHIRPS dataset is adequate to be used for drought risk assessment in case of severe data scarcity for the Bolivian Altiplano. Furthermore, we

found that the vegetation variance can be significantly explained by precipitation and air temperature. More specifically, we point out the relevance of precipitation as the main water source for vegetation development and air temperature as a driver of photosynthetic processes. Precipitation is particularly important at the early and late phenological stages, in which crops are more sensitive to water shortage. This is the case for the main crops in region, i.e., quinoa and potato. For the quinoa crop, the most sensitive phases to water stress are the emergence,

flowering, and grain development (see Geerts et al., 2008; Geerts et al., 2009). The most sensitive phases of the potato crop to water stress is the tuber initiation and bulking (van Loon, 1981; Alva et al., 2012). On the other hand, air temperature is relevant for vegetation productivity, and overall, we found a positive relation between vegetation and air temperature. However, in prolonged dry periods, high air temperature could increase the evapotranspiration rates, and in consequence, decrease the soil moisture (Huang et al., 2019). This scenario could

impact negatively the vegetation, as this is the case of the drought events of 1982-83 and 1997-98, where large production losses were reported (Santos, 2006).

## 4.5. Discussion and Conclusion

We employed a satellite-based and gridded dataset products and tested its empirical accuracy as well as performance to similar (but with coarser resolution) datasets available for the Bolivian Altiplano region.

SAfterwards spatio-temporal patterns of satellite precipitation and gridded air temperature anomalies were explored based on monthly time series during the period of September 1981 to August 2015. Drought severity was evaluated based on a drought classification scheme using NDVI and LST, this classification was related with the ENSO anomalies. Finally, association between the spatial distribution of NDVI with precipitation and air temperature was examined. Using these datasets, it was shown that drought risk severity (measured through

various drought indices) increases substantially during El Niño years (Table 4 and 5), and as a consequence the socio-economic ~~vulnerability~~ drought risk of farmers will likely increase during such periods. ENSO forecasts as well as drought severity (through drought indices) can help to determine possible hotspots of crop deficits during the growing season. ~~Through e~~The empirical relationships ~~with climate variables~~of land surface and climate data

on the local scale our approach can ~~enable~~ support a pro-active approach to disaster risk management against droughts, through an evaluation of the evolution of climate anomalies (in this case the ENSO) and its potential adverse effects in the region. As it was shown here, the ENSO warm phase related characteristics are especially important in the context of extreme drought events and could therefore be incorporated within early warning systems as standard practice. Despite these challenges for development of drought early warning systems (see

FAO, 2016, 2017), applications have been successful in the past (e.g., Global Information and Early Warning System (GIEWS) of FAO, and Famine Early Warning System (FEWS) of USAID). Monitoring and predicting ENSO can therefore significantly contribute to reduce the risk of disasters. This study is a first attempt to provide an assessment of drought impact on agriculture in relation to the ENSO phenomenon for the Bolivian Altiplano. We focused on where vegetation is more affected by droughts over agricultural land and how this can be clarified

using satellite imagery. It is important to note that the variance of drought indices (as well as NDVI) to a large extend is explained by precipitation and air temperature anomalies in the studied region. The agriculture in this semi-arid region is ecologically fragile and the main water source is precipitation, and thus crop production is considerably affected by precipitation anomalies. However, while an overall response of vegetation variance to precipitation and air temperature is evident, it is important to consider other variables, such as evapotranspiration

and soil moisture to improve risk-based models. Another important issue is the time-lag of the response of vegetation to precipitation and air temperature anomalies, which shows a hysteresis of 1-2 months. These findings provide information for future drought risk management and early warning system applications. In addition, with such information agricultural models can be set up and risk management plans with better accuracy determined.

**Acknowledgements**

The authors want to thank the reviewers for their thoughtful comments and efforts towards improving our manuscript. Special thanks to ~~the staff at~~ the International Institute for Applied System Analysis (IIASA), in particular the Young Scientist Summer Program (YSSP) 2017, where this study was initiated~~conceived~~. This research was supported by the Swedish International Development Cooperation Agency (SIDA), and the FORMAS Research Council for Environment, Landscaping and Urban Development. The authors would like to

express their gratitude the Servicio Nacional de Meteorología e Hidrología (SENAMHI) for providing the meteorological data. The authors would also like to thank Ramiro Pillco Zolá and Ángel Aliaga Rivera for the coordination of the research project with the Universidad Mayor de San Andrés of Bolivia.

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

**APPENDIX**

**Table A1.** Spatial location of the studied weather stations where gauged precipitation data are available, the stations that also present temperature maximum and minimum data indicate T on the column of temperature.

| No | Station name | Latitude | Longitude | Altitude | Temperature |
|---|---|---|---|---|---|
| [1] | Ayo Ayo | -17.1 | -68.0 | 3888 | |
| [2] | Calacoto | -17.3 | -68.6 | 3830 | T |
| [3] | Collana | -16.9 | -68.3 | 3911 | T |
| [4] | El Alto Aeropuerto | -16.5 | -68.2 | 4034 | T |
| [5] | El Belen | -16.0 | -68.7 | 3833 | T |
| [6] | Oruro Aeropuerto | -18.0 | -67.1 | 3701 | T |
| [7] | Patacamaya | -17.2 | -67.9 | 3793 | |
| [8] | Salla | -17.2 | -67.6 | 3500 | |
| [9] | San Juan Huancollo | -16.6 | -68.9 | 3829 | |
| [10] | Santiago de Huata | -16.1 | -68.8 | 3845 | T |
| [11] | Tiahuanacu | -16.6 | -68.7 | 3863 | T |
| [12] | Viacha | -16.7 | -68.3 | 3850 | T |

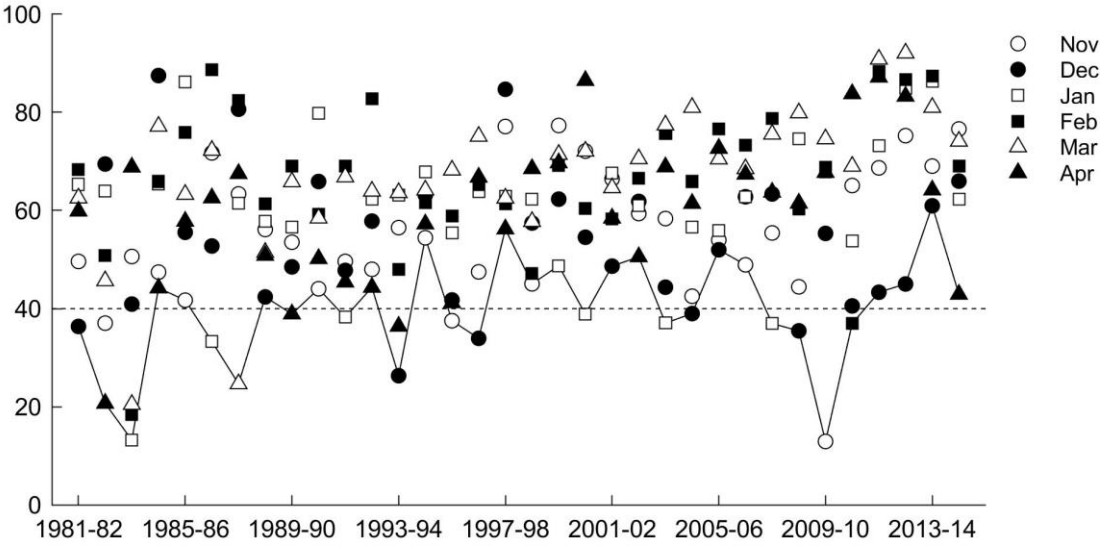

**Fig. A1.** Monthly mean VCI (%) from November 1981 to April 2015. Values below 40% (dashed line) represent a drought event.

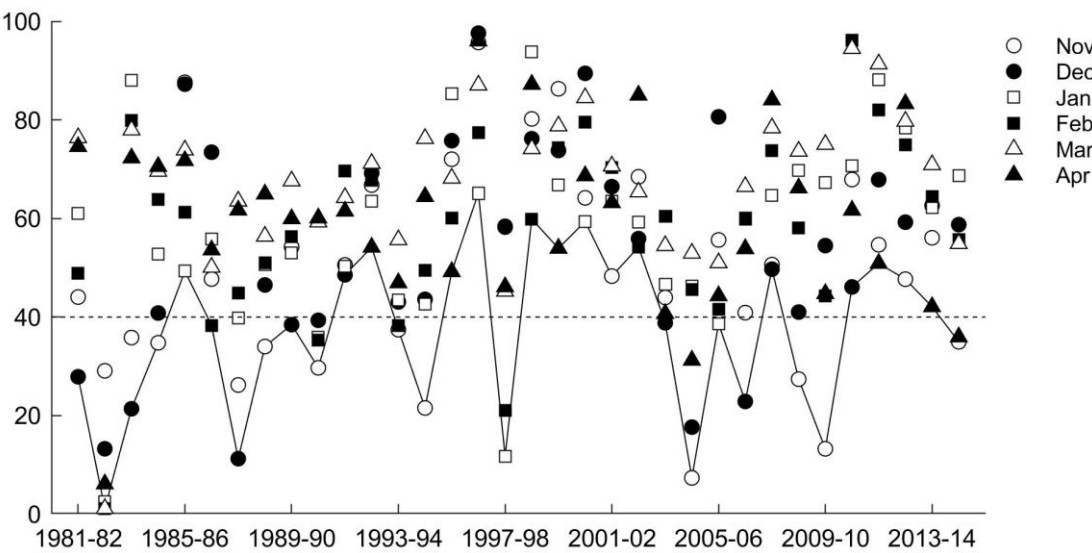

**Fig. A2.** Same as Fig. A1 but for the TCI.

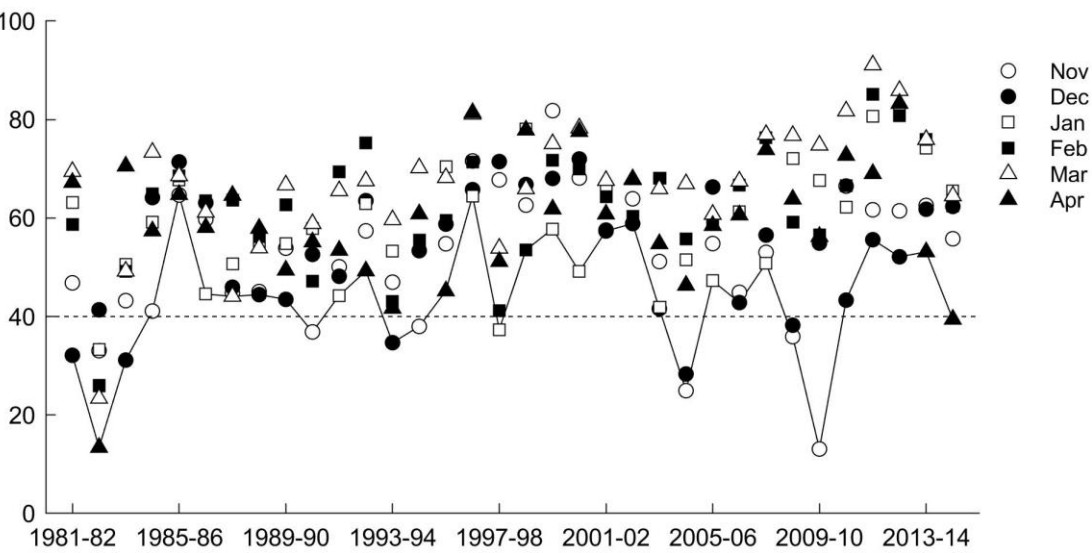

**Fig. A3.** Same as Fig. A1 but for the VHI.