# Peer review of "Drought impact in the Bolivian Altiplano agriculture associated with El Niño—Southern Oscillation using satellite imagery data"

_Natural Hazards and Earth System Sciences, 2018_

## Referee Comment (RC1) · Anonymous Referee #1 · 7 Aug 2019

**General comment**

The authors integrate remote sensing products (Normalized Difference Vegetation Index, land surface temperature, and precipitation), meteorological observations (near-surface air temperature and precipitation), and crop yield data to assess the impacts of ENSO on quinoa and potato yield in the Bolivian Altiplano. The purpose of the study is to develop a statistical framework that can be employed to reduce drought impacts on agricultural production in a region where surface data are scarce. The study shows that the remote sensing products listed above are sufficiently accurate when compared against ground observations, and that the positive ENSO phase significantly decreases

crop yields. The framework is then employed to identity hotspots that are most vulnerable to droughts. The MS presents a relevant contribution to drought-related risk assessments in a region that is poorly studied. My main concern is related to the bias correction of land surface temperature, as explained in the main comments below. Also, the presentation of the methods section requires some attention. I recommend considering the MS for publication in NHESS after major revision.

**Main comments**

- The authors assume that land surface temperature (LST) and near-surface air temperature should be equal. This is a misconception as both variables present different processes. LST directly follows from the Stefan-Boltzmann law and therefore depends on outgoing long wave radiation and surface emissivity. Near-surface air temperature, on the other hand, is affected by other processes, such as turbulent heat fluxes. The authors use near-surface air temperature measurements to "bias correct" remotely-sensed LST using an approach by Zhou and Wang (2016). This does not make much sense, as LST and near-surface air temperature should differ. Furthermore, the cited study by Zhou and Wang (2016) actually uses ground measurements of LST rather than near-surface air temperature to bias correct remotely-sensed LST. I propose three alternative approaches to address this issue: the authors could $(i)$ rerun their analysis using LST directly, $(ii)$ find an approach how to spatially interpolate near-surface air temperature, or $(iii)$ use an already existing air temperature data set that has been published elsewhere (e.g. data from the climate research unit).

- I suggest that the authors improve the presentation of the methods section by including the equations employed in their statistical framework (e.g. the Nash-Sutcliffe efficiency (E) coefficient, POD, and FAR).

**Detailed comments**

P01L14 Please spell out ENSO before using the acronym.

P01L17 You write that "droughts can be better predicted using a combination of satel-
lite imagery and ground-based available data". Better than ground-based avail-
able data alone? Please be explicit.

P01L18 You write that "satellite climate data were associated with" NDVI. This is a
very vague formulation to outline your approach. Please be more precise.

P01L19 You started out your abstract on the topic of drought and are now jumping to
"the crop production variability". Please find a more elegant way to include the
topic of crop production variability. I would include this above when you describe
the research problem.

P01L19 You are jumping back and forth between methods and results. I think you
could improve the readability of your abstract when you first outline your approach
and then the results.

P01L21 I would replace "indicate" with "identify".

P02L02 I would include a reference here, e.g. UNDP, 2011: Tras las huellas del
cambio climatico en Bolivia: Estado del arte del conocimiento sobre adaptacion
al cambio climatico agua y seguridad alimentaria. United Nations Development
Program - Bolivia, 144 pp

P03L14 You could include a reference for SAMS here, e.g. Zhou, J., and K. M. Lau,
1998: Does a monsoon climate exist over South America? J. Climate, 11, 1020-
1040.

P03L19 Please explain more clearly how exactly the gap filling was done.

P03L25 You mention the resolution three times. Please avoid redundancy.

P04L14 Reformulate. I suggest you write "An $E$ equal to 1 corresponds ..."

P04L08 I suggest you provide the equations for the Nash-Sutcliffe efficiency (E) coefficient, POD, and FAR.

P06L10 This paragraph suggests that land surface temperature (LST) and near-surface air temperature should be equal. Please refer to my general comment above to address this misconception.

P06L28 Delete "and the" or restructure sentence.

P07L16 This sentence is vague. Do you mean NDVI grid cells? Also, NDVI does not "simulate" crop yield. Please rephrase.

P07L20 Please define accumulated degree days.

P07L22 Better than what?

P07L26 Spell out and define GDD here.

P11L28 Please refer to my general comment above.

P11L30 Typo, replace p = 001 with p = 0.01.

P12L03 Please refer to my general comment above.

P15L18 I would move any discussion on insurance policy and drought mitigation to the discussion section.

P16L15 Avoid vague formulations such as "There are numerous cases in many countries". Also, it is not accurate to say that the impacts of ENSO are particularly strong in the mid-latitudes.

**Figures**

Figure 01 please specify the percentiles, min, max, and outliers of the boxplots in the Figure caption. The same comment applies to Figure 5.

---

## Referee Comment (RC2) · Anonymous Referee #2 · 12 Oct 2019

GENERAL COMMENT

The paper is focused in the study of drought risk generated by climatic variables during ENSO occurrences and it is oriented to agricultural issues and related impacts on Bolivian Andes. For the last, the authors used potato and quinoa crop measurement data, to be related with temperature and precipitation information on ENSO composite periods. Additionally, the document assessed the detection of specific drought hotspot areas in base the NDVI vegetation index. Crops were related with NDVI variability, and the last was linked with climate variables as precipitation and accumulated degree day data. In general, the document is oriented to impacts on agriculture generate by

droughts during strong El Niño events.

MAIN COMMENTS

The authors didn't clarify their risk definition, for example, in front to an extreme drought even during any kind of warm ENSO phase, the risk can be very low or cero if the direct affected population has very low vulnerability. Then, the mention of risk implies knowledge about the conception of risk, vulnerability and hazardous events (i.e., the danger amount), which are not well described in the current document.

lack of good bibliography review.

P1 section 1. The general idea is the impacts of ENSO in agriculture and food security, but there is not so much to risk.

P3 L4. The title is covering a lot of issues. Risk is not only studied on agricultural context. My suggestion is to change the title to something like "Agricultural drought impacts during the ENSO over the Bolivian Altiplano".

P3 L23. CHIRPS is a good dataset for precipitation information, since it is a mixed observation product (satellite products, station data, etc.), but here is necessary to indicate the problems using it over Andes or over South America. Several papers are pointing out that the CHIRPS across the Andes overestimate/underestimate in lower/higher values, respectively.

Paredes-Trejo et al. 2016. Intercomparison of improved satellite rainfall estimation with CHIRPS gridded product and rain gauge data over Venezuela. https://doi.org/10.20937/ATM.2016.29.04.04

Paredes-Trejo et al. 2017. Validating CHIRPS-based satellite precipitation estimates in Northeast Brazil. https://doi.org/10.1016/j.jaridenv.2016.12.009

Rivera et al. 2018. Validation of CHIRPS precipitation dataset along the Central Andes of Argentina. https://doi.org/10.1016/j.atmosres.2018.06.023

P4 L3. The LST-NDVI association is usually used for drought monitoring, then why didn't the authors explain nothing about it in the introduction and/or in the section 2.1?.

Karnieli et al., 2010. Use of NDVI and Land Surface Temperature for Drought Assessment: Merits and Limitations. https://doi.org/10.1175/2009JCLI2900.1

P5 L5. Since the raw data have cyclicity/periodicity parts, then the 0.7 Spearman correlation should represent a very low association or linearity. Before to start the comparison, it is necessary that the authors remove the cyclicity/periodicity parts from the assessed information.

P6 L10. The LST definition is different that the gauged air temperature from weather stations. LST is defined by Stephan-Boltzmann law, and on the other hand, the air temperature is defined by climate patterns and process. Moreover, as before indicated, the LST-NDVI relationship is a good method for monitoring drought, more than air temperature – NDVI. The authors should work with the LST but need to improve the correction procedure with some in ground LST measurements or other alternative way.

P7 L8. The crop yield vs NDVI is given values on 0.6 Spearman correlation, and it is yielding a little ambiguous result, the authors should bring information like, for example, how much is the explained variance of this relationship? i.e., How much does the NDVI explain the yield?

P7 section 2.4. Was only a set of 2 predictors that were assessed in the regression analysis? if not, which are the other discarded predictors in the regression analysis? more than see the statistical results, the Authors should explain the physical reasons why the others preselected predictors were considered as potential predictors and why they were discarded.

P9 section 3. The data analysis should be done after to remove the cyclicity/periodicity of the data, to be comparable between them.

P10 L7. This could be moved to conclusion section.

[Figure]

P10 L8. "all dataset had acceptable bias" this affirmation is something subjective since the bias can be between 15% to 35%, then it is far to be considered as an acceptable bias. More than the references indicated for the authors (those can show values acceptable in other context), Can the authors show any way or calculation to corroborate that that range of bias is "acceptable"? Another option, in my point of view, is removed this assumption.

P11 L27. Again, the LST temperature has a different physical definition than air temperature. Moreover, the LST- ENSO relationship is given as the ENSO alters the air temperature patterns globally, and that air temperature influences vegetation and agricultural productivity (Glennie and Anyamba, 2018), then on ground level, additionally that air temperature, the vegetation cover, albedo, and soil properties (and others) are affecting the ground temperature generated by emitted radiation on the ground. This means that the ENSO-LST and ENSO- air temperature teleconnections have different mechanisms, then the correction of LST with air temperature has not sense since we expect to assess the crop yields. Hence, the suggestion that the LST underestimation could be due to elevation and/or cloud cover is not correct too.

Glennie and Anyamba, 2018. Midwest agriculture and ENSO: A comparison of AVHRR NDVI3g data and crop yields in the United States Corn Belt from 1982 to 2014. https://doi.org/10.1016/j.jag.2017.12.011

P12 L25-L26. This phrase is ambiguous. Something that the authors can do is to calculate the explained variance per each predictor, and it associates with location coordinates.

P13 L5-L6. Although the lag values are expected to be between 3 or 4 months, the lag differences between precipitation and vegetation per location can be explained on base to local landscape elements (e.g., Yarleque et al. 2016).

Yarleque, C., M. Vuille, D. R. Hardy, A. Posadas, and R. Quiroz (2016), Multiscale assessment of spatial precipitation variability over complex mountain terrain using a

high-resolution spatiotemporal wavelet reconstruction method, J. Geophys. Res. Atmos., 121, 12,198–12,216, doi:10.1002/2016JD025647.

P13 L11-L12. "The hours of sun required for crop development could be the explanation for these results" It is true in part, see me previous comment. On this Andes region is necessary consider aquifer or ground water level changes (i.e., moisture on ground level) from Mountainous regions to flatter/lower elevation areas.

P14 L4. Here was linked a generated index with sea surface temperature anomalies against the crop yield signal with anomalies+ periodicity/cyclicity?. If this is the case, then I expect that the results bring a kind of non-physical statistical information.

Figure 5. In this figure is given boxplots with only the 1982-1983 strong El Niño case as outlier, the rest of cases for quinoa and potato are given a non-statistical difference with other years, since the rest of cases are intercepting the range of the boxplots, i.e., between the maximum and minimum possible values, contradicting the conclusions of the authors.

P16 L1. How is the "magnitude of assistance" calculated/estimated?.

DETAILED COMMENTS

P6 L7. Four or three?

P6 L7. "but not the satellite not and" changes to "but not the satellite and".

P8 L5. "potato was 4°C and 3°C for quinoa" changes to "potato and quinoa were 4°C and 3°C, respectively"

P8 L10. What's "5 percent level" exactly mean?

P9 L7. "with" changes to "during".

P9 L11. Add "strong" before that "El Niño"

P9 L12. Add "strong" before that "El Niño"

P12 L7. Remove "is".

P12 L11. ". And" Changes to ", and".

P14 L7. "warm" changes to "strong".

P15 L2. Add "strong" before that "El Niño".

P15 L9. Add "strong" before that "El Niño".

P15 L20. Remove "is".
* * *

---

## Author Comment (AC1) · 29 May 2020

We want to thank the reviewer for his continued support, detailed comments, and valuable recommendations. Each suggestions was discussed in detail and we have rewritten the paper accordingly. We have also improved the clarity and correctness of phrasing throughout our manuscript. Please find our detailed responses to the reviewer comments below.

General comment The authors integrate remote sensing products (Normalized Difference Vegetation Index, land surface temperature, and precipitation), meteorological observations (nearsurface air temperature and precipitation), and crop yield data to

[Figure]

assess the impacts of ENSO on quinoa and potato yield in the Bolivian Altiplano. The purpose of the study is to develop a statistical framework that can be employed to reduce drought impacts on agricultural production in a region where surface data are scarce. The study shows that the remote sensing products listed above are sufficiently accurate when compared against ground observations, and that the positive ENSO phase significantly decreases crop yields. The framework is then employed to identity hotspots that are most vulnerable to droughts. The MS presents a relevant contribution to drought-related risk assessments in a region that is poorly studied. My main concern is related to the bias correction of land surface temperature, as explained in the main comments below. Also, the presentation of the methods section requires some attention. I recommend considering the MS for publication in NHESS after major revision.

Main comments

- The authors assume that land surface temperature (LST) and near-surface air temperature should be equal. This is a misconception as both variables present different processes. LST directly follows from the Stefan-Boltzmann law and therefore depends on outgoing long wave radiation and surface emissivity. Nearsurface air temperature, on the other hand, is affected by other processes, such as turbulent heat fluxes. The authors use near-surface air temperature measurements to "bias correct" remotely-sensed LST using an approach by Zhou and Wang (2016). This does not make much sense, as LST and near-surface air temperature should differ. Furthermore, the cited study by Zhou and Wang (2016) actually uses ground measurements of LST rather than near-surface air temperature to bias correct remotely-sensed LST. I propose three alternative approaches to address this issue: the authors could (i) rerun their analysis using LST directly, (ii) find an approach how to spatially interpolate near-surface air temperature, or (iii) use an already existing air temperature data set that has been published elsewhere (e.g. data from the climate research unit).

Response: The database used previously in our manuscript was "global monthly

land surface air temperature" from the Global Historical Climatology Network and the Climate Anomaly Monitoring System (GHCN and CAMS) defined by Fan and van den Dool (2008). In the revised version we now used the monthly air temperature dataset from University of Delaware developed by Willmott and Matsuura (see http://climate.geog.udel.edu/~climate/html_pages/README.ghcn_ts2.html). Furthermore, the air temperature database is now properly named along the manuscript.

- I suggest that the authors improve the presentation of the methods section by including the equations employed in their statistical framework (e.g. the NashSutcliffe efficiency (E) coefficient, POD, and FAR).

Response: The equations used for the statistical analysis were now included in the methods section and explicitly referred to throughout the manuscript.

Detailed comments

- P01L14 Please spell out ENSO before using the acronym.

Response: The El Niño Southern Oscillation (ENSO) is now spelled out at the first time that it is mentioned in the manuscript.

- P01L17 You write that "droughts can be better predicted using a combination of satellite imagery and ground-based available data". Better than ground-based available data alone? Please be explicit.

Response: The new version of the manuscript mentions: "The results show that droughts can be monitored using satellite imagery data when ground data are scarce or of poor data quality".

- P01L18 You write that "satellite climate data were associated with" NDVI. This is a very vague formulation to outline your approach. Please be more precise

Response: The manuscript was modified and the new text includes: "...we tested the performance of satellite imagery products for providing vegetation, land surface

temperature (LST), precipitation and air temperature data. With this information, we assessed drought impact on agriculture by associating vegetation with precipitation and air temperature".

- P01L19 You started out your abstract on the topic of drought and are now jumping to "the crop production variability". Please find a more elegant way to include the topic of crop production variability. I would include this above when you describe the research problem.

Response: Now two main modifications were included in the manuscript to avoid confusion. Firstly, the title was modified to: "Drought impact in the Bolivian Altiplano agriculture associated with El Niño Southern Oscillation using satellite imagery data." This includes agriculture as one of the focal points of our study. Secondly, in the abstract the following text was now included to be more specific about our contribution: "Drought is a major natural hazard in the Bolivian Altiplano that causes large agricultural losses, especially during a positive El Niño-Southern Oscillation (ENSO) phase. However, empirical data for drought assessment purposes in this area are scarce, spatially uneven distributed. Due to these limitations we tested the performance of satellite imagery products for providing vegetation, land surface temperature (LST), precipitation and air temperature data on a local level. With this information, the Normalized Difference Vegetation Index (NDVI) and LST were used to classify drought events, and associated with past ENSO phases. It was found that the most severe drought events generally occur during positive ENSO phase (El Niño years). We found a decrease in vegetation is mainly driven by low precipitation and high temperature rates, and we identify areas where losses will be most pronounced under such conditions. The results show that droughts can be monitored using satellite imagery data when ground data are scarce or of poor data quality. The results can be especially beneficial for emergency response operations and for enabling a pro-active approach to disaster risk management against droughts."

- P01L19 You are jumping back and forth between methods and results. I think you

could improve the readability of your abstract when you first outline your approach and then the results

Response: The abstract was modified, please see our response above.

- P01L21 I would replace "indicate" with "identify".

Response: Identify is now used instead.

- P02L02 I would include a reference here, e.g. UNDP, 2011: Tras las huellas del cambio climatico en Bolivia: Estado del arte del conocimiento sobre adaptacion al cambio climatico agua y seguridad alimentaria. United Nations Development Program - Bolivia, 144 pp

Response: The references UNDP, 2011; Garcia and Alavi, 2018 were now included in the text.

- P03L14 You could include a reference for SAMS here, e.g. Zhou, J., and K. M. Lau, 1998: Does a monsoon climate exist over South America? J. Climate, 11, 1020- 1040.

Response: The references Garreaud et al., 2003; Zhou and Lau, 1998 were included.

- P03L19 Please explain more clearly how exactly the gap filling was done.

Response: Data gaps were no longer filled, only in-situ precipitation and temperature data sets with less than 10% of missing data were considered for the analysis. This analysis was carried out relating the in-situ data with the satellite-based data of precipitation and temperature for pair-wise time series. This is mentioned in the section 2.2 Validation of satellite-based data products. We included more information to avoid confusion.

- P03L25 You mention the resolution three times. Please avoid redundancy

Response: Now, the resolution is mentioned only once.

- P04L14 Reformulate. I suggest you write "An E equal to 1 corresponds ..."

Response: E is no longer used as a statistical measurement as other measures as suggested are now introduced.

- P04L08 I suggest you provide the equations for the Nash-Sutcliffe efficiency (E) coefficient, POD, and FAR

Response: All the equations used for the statistical accuracy measures were included in the revised manuscript. Please see Table 1.

- P06L10 This paragraph suggests that land surface temperature (LST) and nearsurface air temperature should be equal. Please refer to my general comment above to address this misconception.

Response: Air temperature, no LST, was used as a predictor. However, it was wrongly named. Now, we have re-written the text and it is properly named in the manuscript (see also response to the main comments).

- P06L28 Delete "and the" or restructure sentence.

Response: "and the data set spans" was deleted from the text.

- P07L16 This sentence is vague. Do you mean NDVI grid cells? Also, NDVI does not "simulate" crop yield. Please rephrase.

Response: This sentence was removed.

- P07L20 Please define accumulated degree days.

Response: ADD is no longer used as a predictor, and the analysis now includes the 3-month time series of air temperature during the growing season instead. We explain the reasoning for that in more detail in the text.

- P07L22 Better than what?

Response: The text was modified to: "For this, only the NDVI grids at the agricultural land were selected".
- P07L26 Spell out and define GDD here.

Response: GDD is no longer used in the analysis.

- P11L28 Please refer to my general comment above.

Response: Please see the main comment response.

- P11L30 Typo, replace p = 001 with p = 0.01.

Response: The typo was corrected.

- P12L03 Please refer to my general comment above.

Response: Please see the main comment response.

- P15L18 I would move any discussion on insurance policy and drought mitigation to the discussion section.

Response: This information was moved to discussion section.

- P16L15 Avoid vague formulations such as "There are numerous cases in many countries". Also, it is not accurate to say that the impacts of ENSO are particularly strong in the mid-latitudes.

Response: This sentence was removed.

- Figure 01 please specify the percentiles, min, max, and outliers of the boxplots in the Figure caption. The same comment applies to Figure 5.

Response: Now included, e.g. lower and upper boundaries 25th (Q1) and 75th (Q3) percentiles, respectively, line inside box is median, lower and upper error lines 1.5 times the interquartile rage (Q3 - Q1) from the top or bottom of the box, white circles data falling outside 1.5 times the interquartile rage.

---

## Author Comment (AC2) · 29 May 2020

We want to thank the reviewer for his continued support, detailed comments, and valuable recommendations. Each suggestions was discussed in detail and we have rewritten the paper accordingly. We have also improved the clarity and correctness of phrasing throughout our manuscript. Please find our detailed responses to the reviewer comments below.

GENERAL COMMENT

The paper is focused in the study of drought risk generated by climatic variables during

ENSO occurrences and it is oriented to agricultural issues and related impacts on Bolivian Andes. For the last, the authors used potato and quinoa crop measurement data, to be related with temperature and precipitation information on ENSO composite periods. Additionally, the document assessed the detection of specific drought hotspot areas in base the NDVI vegetation index. Crops were related with NDVI variability, and the last was linked with climate variables as precipitation and accumulated degree day data. In general, the document is oriented to impacts on agriculture generate by droughts during strong El Niño events.

**MAIN COMMENTS**

- The authors didn't clarify their risk definition, for example, in front to an extreme drought even during any kind of warm ENSO phase, the risk can be very low or cero if the direct affected population has very low vulnerability. Then, the mention of risk implies knowledge about the conception of risk, vulnerability and hazardous events (i.e., the danger amount), which are not well described in the current document.

Response: The reviewed manuscript focuses on drought impact in the Bolivian Altiplano agriculture associated with El Niño Southern Oscillation using satellite imagery data. The aim is to provide information to support disaster risk management using satellite imagery. It is tested/compared to empirical observations so that it can be used for risk reduction of crop production losses. We focus on the severity of drought events. The severity drought is described in the manuscript (please see the results section, Tables 4 and 5, and appendix Fig. A1-A3).

- Lack of good bibliography review.

Response: Previous related studies were reviewed in more detail and relevant information is included in the manuscript, please see reference section.

- P1 section 1. The general idea is the impacts of ENSO in agriculture and food security, but there is not so much to risk
Response: As mentioned above, considering that the manuscript focuses on drought impact on agriculture associated with ENSO. The manuscript title and content now describe more accurately the study approach.

- P3 L4. The title is covering a lot of issues. Risk is not only studied on agricultural context. My suggestion is to change the title to something like "Agricultural drought impacts during the ENSO over the Bolivian Altiplano".

Response: Thank you for the suggestion. The title was modified to "Drought impact in the Bolivian Altiplano agriculture associated with El Niño Southern Oscillation using satellite imagery data"

- P3 L23. CHIRPS is a good dataset for precipitation information, since it is a mixed observation product (satellite products, station data, etc.), but here is necessary to indicate the problems using it over Andes or over South America. Several papers are pointing out that the CHIRPS across the Andes overestimate/underestimate in lower/higher values, respectively.

Paredes-Trejo et al. 2016. Intercomparison of improved satellite rainfall estimation with CHIRPS gridded product and rain gauge data over Venezuela. https://doi.org/10.20937/ATM.2016.29.04.04

Paredes-Trejo et al. 2017. Validating CHIRPS-based satellite precipitation estimates in Northeast Brazil. https://doi.org/10.1016/j.jaridenv.2016.12.009

Rivera et al. 2018. Validation of CHIRPS precipitation dataset along the Central Andes of Argentina. https://doi.org/10.1016/j.atmosres.2018.06.023

Response: Thank you for the references. They were very helpful. The manuscript now indicates the uncertainties of using satellite-based precipitation data, and the recommended references are included in the results section.

- P4 L3. The LST-NDVI association is usually used for drought monitoring, then why didn't the authors explain nothing about it in the introduction and/or in the section 2.1?
Karnieli et al., 2010. Use of NDVI and Land Surface Temperature for Drought Assessment: Merits and Limitations. https://doi.org/10.1175/2009JCLI2900.1

Response: The manuscript now includes more information about NDVI and LST as relevant drought indicators. Moreover, the classification of drought using NDVI and LST is now included in detail as well (see sections 2.3 and 3).

- P5 L5. Since the raw data have cyclicity/periodicity parts, then the 0.7 Spearman correlation should represent a very low association or linearity. Before to start the comparison, it is necessary that the authors remove the cyclicity/periodicity parts from the assessed information.

Response: To avoid errors from periodicity, the accuracy measures of the satellitebased data products of precipitation and air temperature were defined for each month of the time series (see sections 2.3 and 3).

- P6 L10. The LST definition is different that the gauged air temperature from weather stations. LST is defined by Stephan-Boltzmann law, and on the other hand, the air temperature is defined by climate patterns and process. Moreover, as before indicated, the LST-NDVI relationship is a good method for monitoring drought, more than air temperature – NDVI. The authors should work with the LST but need to improve the correction procedure with some in ground LST measurements or other alternative way.

Response: The database used previously was "a global monthly land surface air temperature" from the Global Historical Climatology Network and the Climate Anomaly Monitoring System (GHCN and CAMS) defined by Fan and van den Dool (2008). For the modified manuscript, we used the monthly air temperature dataset from University of Delaware developed by Willmott and Matsuura (see http://climate.geog.udel.edu/~climate/html\_pages/README.ghcn\_ts2.html). Now, the air temperature database is properly named along the manuscript.

- P7 L8. The crop yield vs NDVI is given values on 0.6 Spearman correlation, and it is
yielding a little ambiguous result, the authors should bring information like, for example, how much is the explained variance of this relationship? i.e., How much does the NDVI explain the yield?

Response: The reviewed manuscript does not include the association of crop yield and NDVI as a technique to discard NDVI grids. In contrast, we now assume that NDVI generally simulates properly the crop production. This is because the elimination of NDVI grids from the agricultural land could ignore relevant information. As well, we want to avoid some uncertainties originated from the crop yield dataset. For instance, the crop yield data do not take in consideration the crop rotation that are represented by different crops in the same area across sequenced growing seasons. We include the limitations as well as advantages using this approach in the discussion section and also provide some ways forward in that regard.

- P7 section 2.4. Was only a set of 2 predictors that were assessed in the regression analysis? if not, which are the other discarded predictors in the regression analysis? more than see the statistical results, the Authors should explain the physical reasons why the others preselected predictors were considered as potential predictors and why they were discarded.

Response: This text is now included in the manuscript: "For the study, we assumed that NDVI simulates the stages of the crop phenological stages that is from September to April (Fig. 1). Precipitation was selected as predictor for its relevance on water availability for vegetation growth. Precipitation is the main source of water in the Altiplano because only 9 percent of the Bolivian cropped surface area is irrigated (INE, 2015). Air temperature is a relevant variable due to its involvement on photosynthetic and respiration processes (Karnieli et al., 2010)." We also discussed the results now in more detail.

- P9 section 3. The data analysis should be done after to remove the cyclicity/periodicity of the data, to be comparable between them.
Response: To avoid errors originated from cyclicity/periodicity, now the analysis is developed for each month for the accuracy measures of satellite/based data products and the classification of drought. The stepwise regression between NDVI and climate variables were developed using a standardized 3-month time series. "Previous to the stepwise regression analysis, the 3-month time series of NDVI, satellite precipitation and satellite air temperature were standardized".

- P10 L7. This could be moved to conclusion section.

Response: The text was modified (see sections 3 and 4).

- P10 L8. "all dataset had acceptable bias" this affirmation is something subjective since the bias can be between 15% to 35%, then it is far to be considered as an acceptable bias. More than the references indicated for the authors (those can show values acceptable in other context), Can the authors show any way or calculation to corroborate that that range of bias is "acceptable"? Another option, in my point of view, is removed this assumption.

Response: Now the text includes: "Summarizing these observations we conclude that CHIRPS-rainfall dataset is an adequate alternative in case of lack of gauged data or in case of poor data quality. However, it should be noted that such data still must be used with caution considering the uncertainties due to the under or overestimation of precipitation along the heterogeneous topography of the Altiplano (see Paredes-Trejo et al., 2016; Paredes-Trejo et al., 2017; Rivera et al., 2018)."

And:

"In conclusion, the satellite air temperature data product perform adequately from November to April. Similar to the precipitation data, the application of satellite air temperature data must take into account the potential errors due to the estimation uncertainties, mainly during winter season".

- P11 L27. Again, the LST temperature has a different physical definition than air

NHESSD
temperature. Moreover, the LST- ENSO relationship is given as the ENSO alters the air temperature patterns globally, and that air temperature influences vegetation and agricultural productivity (Glennie and Anyamba, 2018), then on ground level, additionally that air temperature, the vegetation cover, albedo, and soil properties (and others) are affecting the ground temperature generated by emitted radiation on the ground. This means that the ENSO-LST and ENSO- air temperature teleconnections have different mechanisms, then the correction of LST with air temperature has not sense since we expect to assess the crop yields. Hence, the suggestion that the LST underestimation could be due to elevation and/or cloud cover is not correct too. Glennie and Anyamba, 2018. Midwest agriculture and ENSO: A comparison of AVHRR NDVI3g data and crop yields in the United States Corn Belt from 1982 to 2014. https://doi.org/10.1016/j.jag.2017.12.011

Response: As mentioned above the database used in the analysis was air temperature, however it was misnamed, now it is properly named along the text. Moreover, now we employed another air temperature data base that is the monthly air temperature dataset from University of Delaware developed by Willmott and Matsuura (http://climate.geog.udel.edu/~climate/html\_pages/README.ghcn\_ts2.html).

- P12 L25-L26. This phrase is ambiguous. Something that the authors can do is to calculate the explained variance per each predictor, and it associates with location coordinates.

Response: This text was removed, and now the results show the findings using the spatial coordinates (see Fig. 5).

- P13 L5-L6. Although the lag values are expected to be between 3 or 4 months, the lag differences between precipitation and vegetation per location can be explained on base to local landscape elements (e.g., Yarleque et al. 2016). Yarleque, C., M. Vuille, D. R. Hardy, A. Posadas, and R. Quiroz (2016), Multiscale assessment of spatial precipitation variability over complex mountain terrain using a high-resolution spatiotem-
poral wavelet reconstruction method, J. Geophys. Res. Atmos., 121, 12,198–12,216, doi:10.1002/2016JD025647.

Response: Thank you for the useful reference, it is now included in the results section of the manuscript.

- P13 L11-L12. "The hours of sun required for crop development could be the explanation for these results" It is true in part, see me previous comment. On this Andes region is necessary consider aquifer or ground water level changes (i.e., moisture on ground level) from Mountainous regions to flatter/lower elevation areas.

Response: The manuscript now mentions the findings of Yarleque et al. (2016).

- P14 L4. Here was linked a generated index with sea surface temperature anomalies against the crop yield signal with anomalies+ periodicity/cyclicity?. If this is the case, then I expect that the results bring a kind of non-physical statistical information.

Response: Now the analysis includes the classification of drought using NDVI and LST. The drought events were analyzed and compared with ENSO phases. The classification of drought was developed for each month to avoid errors from periodicity.

- Figure 5. In this figure is given boxplots with only the 1982-1983 strong El Niño case as outlier, the rest of cases for quinoa and potato are given a non-statistical difference with other years, since the rest of cases are intercepting the range of the boxplots, i.e., between the maximum and minimum possible values, contradicting the conclusions of the authors.

Response: This figure is no longer in the manuscript. More information to avoid confusion in regards to results found was included.

- P16 L1. How is the "magnitude of assistance" calculated/estimated?

Response: This sentence was modified to "Our approach can enable a pro-active approach to disaster risk management against droughts."
**DETAILED COMMENTS**

- P6 L7. Four or three? P6 L7. "but not the satellite not and" changes to "but not the satellite and". P8 L5. "potato was 4âŮęC and 3âŮęC for quinoa" changes to "potato and quinoa were 4âŮęC and 3âŮęC, respectively" P8 L10. What's "5 percent level" exactly mean? P9 L7. "with" changes to "during". P9 L11. Add "strong" before that "El Niño" P9 L12. Add "strong" before that "El Niño". P12 L7. Remove "is". P12 L11. ". And" Changes to ", and". P14 L7. "warm" changes to "strong". P15 L2. Add "strong" before that "El Niño". P15 L2. Add "strong" before that "El Niño". P15 L2. Remove "is".

Response: All the detailed comments from page 6 to page 15 were modified following the referee suggestions.

**NHESSD**

---

## Referee Report (RR1)

MS Title: Drought risk in the Bolivian Altiplano associated with El Nino Southern Oscillation using satellite imagery data

Authors: M. Pontoppidan, E. W. Kolstad, S. P. Sobolowski, A. Sorteberg, C. Liu, R. Rasmussen

**General comments**

- The main comment in my first review was that the authors used near-surface air temperature measurements to "bias correct" remotely-sensed land surface temperature. This approach is not acceptable, as land surface temperature is not the same as near-surface air temperature, which I explained in my previous review in more detail. The authors have responded to this comment as follows:

  "The database used previously in our manuscript was global monthly land surface air temperature from the Global Historical Climatology Network and the Climate Anomaly Monitoring System (GHCN and CAMS) defined by Fan and van den Dool (2008). In the revised version we now used the monthly air temperature dataset from University of Delaware developed by Willmott and Matsuura."

  The respective data set is not a remote sensing product, but a globally gridded data set that is based on the spatial interpolation of in situ-measurements[1]. However, the authors characterize this data set as a "satellite air temperature dataset" (page 4, line 4), which is false.

- Furthermore, the land surface temperature data set used in the MS was obtained from the Global Land Data Assimilation System (GLDAS) by the Noah Land Surface Model L4 monthly version 2.0. As the name suggests, this is not remotely sensed land surface temperature, but outputs from a land surface model that is forced with observations, including remotely sensed data. However, the authors claim this data to be a "satellite imagery product" (see abstract). Why was this data set chosen? The authors could have used remotely-sensed LST from AVHRR or MODIS, instead.

- Also, the abstract states that the authors "tested the performance of satellite imagery products for providing vegetation, land surface temperature (LST), precipitation and air temperature data on a local level". To my understanding, the MS only evaluates air temperature and precipitation against in situ measurement, not NDVI or LST.

- Please correct these misrepresentations. Also include a description of the gridded air temperature and LST data used in your study and justify why you chose those products.

**Minor comments**

P01L18 "With *this* information ...". Vague formulation. Be more explicit

P06L05 "These measures were used to evaluate the satellite estimations" Which variables?
* * *
[1] https://psl.noaa.gov/data/gridded/data.UDel_AirT_Precip.html

P06L05 Temperature biases are usually assessed in absolute rather than relative terms. Please replace the relative bias and MAE in Figure 4 with absolute values.

P07L10 "Healthy vegetation [...] shows a low surface temperature due to the absorption of thermal infrared radiation". This does not make any physical sense.

P09L04 "Precipitation is the main source of water in the Altiplano because only 9% of the Bolivian cropped surface area are irrigated". What is with the supply of water from melting glaciers?

---

## Referee Report (RR2)

**Review of "Drought impact in the Bolivian Altiplano agriculture associated with El Nino Southern Oscillation using satellite imagery data"**

The authors of this manuscript blend a variety of in situ and satellite-based data to assess the impact of drought on crop yield in the Bolivian Altiplano. Overall, I found the paper to be well written, with excellent figures. However, I think the paper lacks direction, with the aims and objectives rather unclear. Looking at the conclusion, the main outcome of the paper is to demonstrate the link between ENSO and agricultural drought in the Bolivian altiplano, but it is not apparent from the introduction that this will be the main subject of the work. Overall, I think a more comprehensive introduction is needed both to establish the main purpose of the manuscript, and to place the work in the context of other such efforts to use satellite-based data for drought risk management. As other reviewers have pointed out, the authors could use the introduction to establish a theoretical framework of risk (e.g. risk = hazard * vulnerability * exposure) and drought (meteorological, agricultural, and so on). Readers unfamiliar with this part of the world may also benefit from a discussion of the ENSO phenomenon, the mechanisms through which it may cause droughts, and the challenges of predicting its likely effects.

Section 2.1 (Ground data and satellite imagery) would make more sense as two sections describing (i) meteorological data; and (ii) land surface data (i.e. NDVI and LST). Additionally, I would suggest separating the methods section, and providing a short overview at the start of this new section of the overall approach.

The satellite air temperature data used in this study has a resolution of 0.5 degree, which is relatively coarse compared to other input data, especially considering the variability in elevation in the altiplano. Could the authors clarify whether this data was downscaled using a lapse rate?

MODIS data would provide LST and NDVI at a much higher spatial resolution (500m versus 1/12 degree) but for a shorter period (2000-present versus 1981-2015). The authors should justify their preference for temporal coverage over spatial resolution. My concern here would be that at such a coarse resolution the agricultural signal may be much smaller than the signal from natural vegetation, and hence the analysis may not adequately account for aspects of on-farm water management such as irrigation, mulching, and crop selection – or indeed not growing crops at all and seeking off-farm work. Perhaps this isn't an issue for this study; regardless, the authors should allay the readers concerns.

To add weight to the argument presented by the manuscript, perhaps the authors could consider showing yield data for the study period for the crops in question? This would go some way to establishing the link between drought and agricultural risk.

In the conclusion, lines 7-9, I think the authors may be confusing risk with hazard exposure, and socio-economic vulnerability with risk. This is one reason for establishing the theoretical framework for risk/hazard/vulnerability/exposure at the outset (see previous comment above).

In the Conclusion, the authors state that "Through empirical research with climate variables on the local scale our approach can enable a proactive approach to disaster risk management against droughts." I think this requires elaboration – it's not actually clear how the work in this manuscript could contribute to such a system. The main result of the paper appears to be to show that El Nino years are associated with more severe droughts in the Bolivian Altiplano. This comes back to the lack of clarity about the purpose of the manuscript at the outset.

---

## Author Response (AR2)

**Responses to referee comments:**

**Referee # 2, report # 1:**

The authors want to thank the referee for support of the paper revision.

**Referee #1 report #2**

The authors want to thank the referee for his/her support and detailed comments of the paper revision, and help to significantly improve the manuscript. The authors would like to express the appreciation in the acknowledgments section.

MS Title: Drought risk in the Bolivian Altiplano associated with El Nino Southern Oscillation using satellite imagery data

Authors: M. Pontoppidan, E. W. Kolstad, S. P. Sobolowski, A. Sorteberg, C. Liu, R. Rasmussen

**Answer:** Please note that the authors are Claudia Canedo-Rosso, Stefan Hochrainer-Stigler, Georg Pflug, Bruno Condori, and Ronny Berndtsson.

General comments

- The main comment in my first review was that the authors used near-surface air temperature measurements to "bias correct" remotely-sensed land surface temperature. This approach is not acceptable, as land surface temperature is not the same as near-surface air temperature, which I explained in my previous review in more detail. The authors have responded to this comment as follows:
  "The database used previously in our manuscript was global monthly land surface air temperature from the Global Historical Climatology Network and the Climate Anomaly Monitoring System (GHCN and CAMS) defined by Fan and van den Dool (2008). In the revised version we now used the monthly air temperature dataset from University of Delaware developed by Willmott and Matsuura."

  The respective data set is not a remote sensing product, but a globally gridded data set that is based on the spatial interpolation of in situ-measurements1 . However, the authors characterize this data set as a "satellite air temperature dataset" (page 4, line 4), which is false.

  **Answer:** We agree on this comment and modified the manuscript accordingly, i.e. the air temperature dataset is now described as a gridded time series dataset and is no longer termed a satellite-based dataset.

  Furthermore, the land surface temperature data set used in the MS was obtained from the Global Land Data Assimilation System (GLDAS) by the Noah Land Surface Model L4 monthly version 2.0. As the name suggests, this is not remotely sensed land surface temperature, but outputs from a land surface model that is forced with observations, including remotely sensed data. However, the authors claim this data to be a "satellite imagery product" (see abstract). Why was this data set chosen? The authors could have used remotely-sensed LST from AVHRR or MODIS, instead.

**Answer:** The main reason that GLDAS was chosen was the time period of the LST dataset product that covers the study period. In contrast, MODIS has a better spatial resolution but a shorter time availability (the first MODIS instrument was launched in December 1999, and the second in May 2002). Moreover, LST estimations of GLDAS are defined using remotely sensed observations from AVHRR (see Data assimilation section in Rodell et al, 2004) and include an algorithm that relies on an optimal interpolation routine (Ottlé and Vidal-Madjar, 1992) to assimilating the LST onto a 0.25 to 0.25 degree grid. The manuscript now includes an extended description of GLDAS and the LST dataset.

Rodell, M., Houser, P. R., Jambor, U., Gottschalck, J., Mitchell, K., Meng, C.-J., Arsenault, K., Cosgrove, B., Radakovich, J., Bosilovich, M., Entin, J. K., Walker, J. P., Lohmann, D., and Toll, D.: The Global Land Data Assimilation System, Bull. Amer. Meteor. Soc., 85, 381-394, 10.1175/bams-85-3-381, 2004.

Ottlé, C., and Vidal-Madjar, D.: Estimation of land surface temperature with NOAA9 data, Remote Sens. Environ., 40, 27-41, https://doi.org/10.1016/0034-4257(92)90124-3, 1992.

- Also, the abstract states that the authors "tested the performance of satellite imagery products for providing vegetation, land surface temperature (LST), precipitation and air temperature data on a local level". To my understanding, the MS only evaluates air temperature and precipitation against in situ measurement, not NDVI or LST.

  **Answer:** Thank you for this remark. The manuscript was modified to describe in more detail that the performance evaluation was developed for precipitation and air temperature datasets.

- Please correct these misrepresentations. Also include a description of the gridded air temperature and LST data used in your study and justify why you chose those products.

  **Answer:** We agree and the manuscript was modified along the suggestions of the referee. The justification of the dataset selection was included.

Minor comments

- P01L18 "With this information ...". Vague formulation. Be more explicit
  **Answer:** The manuscript was modified, and "with this information" is no longer included.

- P06L05 "These measures were used to evaluate the satellite estimations" Which variables?
  **Answer:** To clarify the variables that were evaluated the text "Precipitation satellite estimation and gridded air temperature data" was included in the manuscript.

- P06L05 Temperature biases are usually assessed in absolute rather than relative terms. Please replace the relative bias and MAE in Figure 4 with absolute values.
  **Answer:** The absolute values of MAE instead of the relative ones were included in the paper. Please note that mean error (ME, Fig. 4) already shows the absolute values of the temperature record, for this reason bias was kept as relative value.

- P07L10 "Healthy vegetation [...] shows a low surface temperature due to the absorption of thermal infrared radiation". This does not make any physical sense.
  **Answer:** The manuscript text was modified to: Healthy vegetation usually shows enlarged near infrared and reduced visible red band, and emits less absorbed thermal infrared radiation resulting in lower surface temperature (Kogan and Guo, 2017).

- P09L04 "Precipitation is the main source of water in the Altiplano because only 9% of the Bolivian cropped surface area are irrigated". What is with the supply of water from melting glaciers?
  **Answer:** The glacial melt content as water source for irrigation is low in the central Andes. And, it covers about 5% of the total agricultural area, from 398 km$^2$ to 2096 km$^2$ (Buytaert et al., 2017).

  Buytaert, W., Moulds, S., Acosta, L., De Bièvre, B., Olmos, C., Villacis, M., Tovar, C., and Verbist, K. M. J.: Glacial melt content of water use in the tropical Andes, Environmental Research Letters, 12, 114014, 10.1088/1748-9326/aa926c, 2017.

**Referee #3 report #3**

The authors want to thank the reviewer for thoughtful comments and efforts towards improving our manuscript.

- Review of "Drought impact in the Bolivian Altiplano agriculture associated with El Nino Southern Oscillation using satellite imagery data" The authors of this manuscript blend a variety of in situ and satellite-based data to assess the impact of drought on crop yield in the Bolivian Altiplano. Overall, I found the paper to be well written, with excellent figures. However, I think the paper lacks direction, with the aims and objectives rather unclear. Looking at the conclusion, the main outcome of the paper is to demonstrate the link between ENSO and agricultural drought in the Bolivian Altiplano, but it is not apparent from the introduction that this will be the main subject of the work. Overall, I think a more comprehensive introduction is needed both to establish the main purpose of the manuscript, and to place the work in the context of other such efforts to use satellite-based data for drought risk management. As other reviewers have pointed out, the authors could use the introduction to establish a theoretical framework of risk (e.g. risk = hazard * vulnerability * exposure) and drought (meteorological, agricultural, and so on). Readers unfamiliar with this part of the world may also benefit from a discussion of the ENSO phenomenon, the mechanisms through which it may cause droughts, and the challenges of predicting its likely effects.

  **Answer:** We agree and thank him for this suggestion. The aim of the research is now included explicitly in the abstract and introduction section. We now included the definition of disaster risk from the well established IPCC framework and also gave a discussion of possible drought types. We state our research contribution also now more clearly by including the following sentence: "Using these concepts and definitions our research aims to: (1) classify agricultural drought severity (utilizing NDVI as a proxy for crop yields) using land surface data (our exposure component) and climate data (our hazard component), (2) analyse the relationship between drought and ENSO, and (3) assess drought through examining the spatial-temporal variability of vegetation (including the vulnerability component through statistical analysis), and its association with climate data." Furthermore, we added some additional references for the interested reader.

An explanation of the ENSO phenomenon and its effects in the study area were included in the Introduction section. Also, a description of the challenges for the ENSO effects prediction was also included.

- Section 2.1 (Ground data and satellite imagery) would make more sense as two sections describing (i) meteorological data; and (ii) land surface data (i.e. NDVI and LST). Additionally, I would suggest separating the methods section, and providing a short overview at the start of this new section of the overall approach.

**Answer:** The authors agree and data used are now organized in two different subsections in the data section: climate data, and land surface data.

An overall approach was defined at the beginning of the Methodology section:

"The analysis of drought impact on agriculture for the Bolivian Altiplano and its relationship with the ENSO is based on the following three steps. Firstly, an evaluation of satellite precipitation and gridded air temperature against gauged datasets was performed to investigate the accuracy of these estimates compared to empirical on-the-ground date. Secondly, the severity of drought was classified using land surface data, and using this information drought events were associated with the ENSO variability. Finally, a stepwise regression approach was used to study the variability of vegetation and its relationship with corresponding climate variables. The overall aim of our study is to investigate drought effects on agriculture through the analysis of land surface and climate variations, and its relation with the ENSO anomalies."

- The satellite air temperature data used in this study has a resolution of 0.5 degree, which is relatively coarse compared to other input data, especially considering the variability in elevation in the Altiplano. Could the authors clarify whether this data was downscaled using a lapse rate?

**Answer:** The data was not downscaled using a lapse rate, however the used dataset has incorporated station-height information, through an average air-temperature lapse rate (Willmott and Matsuura, 1995). Here, a digital-elevation-model was used for the interpolation to adjust air temperatures in relation to the sea level. A more detailed description about the gridded air temperature data is now included in the manuscript.

Willmott, C. J. and K. Matsuura (1995) Smart Interpolation of Annually Averaged Air Temperature in the United States. Journal of Applied Meteorology, 34, 2577-2586.

- MODIS data would provide LST and NDVI at a much higher spatial resolution (500m versus 1/12 degree) but for a shorter period (2000-present versus 1981-2015). The authors should justify their preference for temporal coverage over spatial resolution. My concern here would be that at such a coarse resolution the agricultural signal may be much smaller than the signal from natural vegetation, and hence the analysis may not adequately account for aspects of on-farm water management such as irrigation, mulching, and crop selection – or indeed not growing crops at all and seeking off-farm work. Perhaps this isn't an issue for this study; regardless, the authors should allay the readers concerns.

**Answer:** The authors agree and have now mentioned in the manuscript (see Data Used section) the uncertainty from the low spatial resolution and a recommendation for improving the analysis by using satellite data with better spatial resolution.

- To add weight to the argument presented by the manuscript, perhaps the authors could consider showing yield data for the study period for the crops in question? This would go some way to establishing the link between drought and agricultural risk.

  **Answer:** The previous manuscript version did included information on agriculture. Here, crop yield of quinoa and potato crops were analyzed. However, due to the coarse resolution (only three records were available over the entire Bolivian Altiplano) and low quality of the agricultural records, the authors decided to exclude the data from the manuscript to avoid the uncertainty that this information could bring in.

- In the conclusion, lines 7-9, I think the authors may be confusing risk with hazard exposure, and socio-economic vulnerability with risk. This is one reason for establishing the theoretical framework for risk/hazard/vulnerability/exposure at the outset (see previous comment above).

  **Answer:** We have reformulated the text to: Using these datasets, it was shown that drought severity (measured through various drought indices) increases substantially during El Niño years (Table 4 and 5), and as a consequence the socio-economic drought risk of farmers will likely increase during such periods.

- In the Conclusion, the authors state that "Through empirical research with climate variables on the local scale our approach can enable a proactive approach to disaster risk management against droughts." I think this requires elaboration – it's not actually clear how the work in this manuscript could contribute to such a system. The main result of the paper appears to be to show that El Nino years are associated with more severe droughts in the Bolivian Altiplano. This comes back to the lack of clarity about the purpose of the manuscript at the outset.

[revised manuscript text omitted]